# A UNIVERSAL METRIC OF DATASET SIMILARITY FOR MULTI-SOURCE LEARNING

## ABSTRACT

Multi-source learning is a machine learning approach that involves training on data from multiple sources. Applied domains such as healthcare and finance have been increasingly using multi-source learning to improve model performance. However, datasets collected from different sources can be non-identically distributed, leading to degradation in model performance. Most existing methods for assessing dataset similarity are limited by being dataset or task-specific. They propose similarity metrics that are either unbounded and dependent on dataset dimension and scale, or require model-training. Moreover, these metrics can only be calculated by exchanging data across sources, which can be a privacy concern in domains such as healthcare and finance. To address these challenges, we propose a novel bounded metric for assessing dataset similarity. Our metric exhibits several desirable properties: it is dataset-agnostic, considers label information, and requires no model training. First, we establish a theoretical connection between our metric and the learning process. Next, we extensively evaluate our metric on a range of real-world datasets and demonstrate that our metric assigns scores that align with how these data were collected. Further, we show a robust and interpretable relationship between our metric and multi-source learning performance. Finally, we provide a privacy-preserving method to calculate our metric. Our metric can provide valuable insights for deep learning practitioners using multi-source datasets.

## 1 INTRODUCTION

Multi-source learning (MSL) encompasses all methods that use datasets collected from different sources for model training. It includes transfer learning, federated learning, meta-learning, and joint-learning Crammer et al. (2006). MSL is increasingly used in deep learning where more complex models require larger datasets for training. A common practical setting for MSL is when multiple institutions collaborate to train a deep learning model (*e.g.,* hospitals, or banks) Huang et al. (2021). In such situations, MSL is a promising approach to increase sample size and dataset diversity, leading to improved model performance and generalizability Zhao et al. (2020). However, when datasets are collected from different sources, they often exhibit distribution shifts, making them non-identically independently distributed (non-IID) Hsieh et al. (2020). This shift can worsen performance and is due to factors that affect either label or feature distributions (*e.g.,* label skew or, concept drift and shift, respectively)Hsieh et al. (2020). The problem of non-IID data is common in many practical settings where datasets capture both the phenomenon of interest as well as other heterogeneous factors contributing to data generation Zhang et al. (2022); Hripcsak & Albers (2013). As the causes of non-IID are varied yet common, it is often challenging to diagnose when and how datasets differ. In practical settings, the reduced performance can have negative real-world implications and limit future collaborations Li et al. (2019); Tu et al. (2022); Yu et al. (2022). Therefore, it is crucial to assess distribution simialrity between datasets before conducting MSL.

Current methods for evaluating dataset similarity have been successful in the field of domain adaptation and transfer learning Alvarez-Melis & Fusi (2020); Courty et al. (2017); Damodaran et al. (2018); Pándy et al. (2021); Cortes & Mohri (2011); Achille et al. (2018; 2019); Ben-David et al. (2006); Mansour et al. (2009). Broadly, these methods fall under three categories: leveraging notions of distance/similarity between datasets, learning a mapping function between datasets, and estimating parameter sensitivity of the models. All of these categories suffer from limitations. Firstly, they are dataset or task specific. In particular, methods that use distance or similarity metrics (*e.g.,*

KL-divergence, Wasserstein distance) give unbounded scores for the similarity between distributions of two datasets Alvarez-Melis & Fusi (2020). This makes interpretation of the score reliant on a thorough understanding of the dataset and task, *i.e.,* the same absolute score can have different interpretations depending on the scale and dimensionality of the data. Furthermore, these metrics do not scale to high dimensional data. Methods that learn a mapping function or estimate parameter sensitivity rely on training a specific model for the task. These methods also require large sample sizes to estimate parameters (*e.g.,* Vapnik-Chervonenkis-dimension or Rademacher complexity). Secondly, most methods are not suited to supervised tasks as few methods, except Alvarez-Melis & Fusi (2020), take into account label distributions. Finally, most methods require access to datasets from different sources, which might violate privacy regulations in domains that use sensitive data to train models (*e.g.,* healthcare) Butler (2007).

We propose a novel bounded metric that measures dataset distribution similarity and identifies when MSL would be beneficial for the task. Our metric addresses some limitations of existing methods. Specifically, it (1) provides a universal measure of distribution shifts between datasets that is not specific to a learning task or dataset; (2) is label-aware; (3) does not require model training; and (4) can be calculated in a privacy-preserving manner. Our metric leverages optimal transport (OT) to compute a bounded score between every pair of data points, taking into account both feature and label distributions. We achieve this by replacing the Wasserstein distance with a hybrid Cosine Similarity-Hellinger distance between data points. We provide a theoretical justification for why this metric can capture model training dynamics. We then demonstrate that our metric correlates with model performance when comparing MSL training to training on a single site. Specifically, we show that our metric yields consistent results across seven datasets of different dimensions and scale and four different MSL learning algorithms. Our approach has the potential to avoid inefficient training on non-IID data, hence improving the performance of MSL.

## 2 CONTRIBUTIONS

In this work, we seek to address the challenge of assessing dataset similarity in MSL, with a focus on supervised learning:

1. We introduce a bounded dataset similarity metric that is dataset agnostic, label-aware, and can be calculated without model training. The metric ranges from 0 to 1, where 0 indicates IID datasets and 1 indicates completely non-IID datasets. Our metric uses OT with a cost function that is hybrid Cosine Similarity-Hellinger distance between pairs of data points.

2. We present a theoretical framework for why a Cosine Similarity-Hellinger distance cost function can evaluate dataset similarity and predict MSL performance. We demonstrate that there is high probability of any pair of data points taken from two random datasets being orthogonal in high dimension and that deep learning models leverage the non-orthogonality of related data points for training. We also show that our dataset similarity metric is correlated to gradient diversity, a measure of task or dataset dissimilarity. Importantly, this indicates that our metric is correctly identifying training dynamics without model training.

3. We investigate the empirical relationship between our dataset similarity metric and training performance across seven datasets and four MSL learning algorithms. Our findings are consistent irrespective of dataset scale and dimension. We find that a score of 0.2 or lower lead to improved learning performance, while scores greater than 0.3 result in negative learning. Further, we show that while loss-based non-IID learning algorithms limit the effect of negative learning at high dataset dissimilarity scores, they also reduce the benefit of MSL at low scores.

4. To address data privacy concerns in distributed settings, we leverage methods from Secure Multi-Party Computation and Differential Privacy to provide a privacy-preserving method for feature and label cost, respectively. We show that our method has high accuracy and low computational overhead, two desirable properties for privacy-preserving methods.

## 3 RELATED WORK

Previous studies on calculating multi-source dataset similarity have primarily addressed problems found in domain adaptation and meta-learning. However, there has been limited focus on learning tasks commonly employed in applied fields such as healthcare and finance. The significance of this

gap becomes particularly evident when considering the unique challenges these domains present, including: (1) the need to incorporate label distributions for supervised learning tasks that are extensively used, (2) the computational demands and task specificity constraints of the methods, (3) the necessity for bounded similarity measures to facilitate interpretation and comparison across datasets and tasks, and (4) the importance of privacy preservation given data sharing constraints, especially in domains like healthcare. None of the existing methods consider privacy preservation and only one of them offers a bounded dataset similarity measure.

Alvarez-Melis & Fusi (2020) proposed a novel OT based approach that incorporated label-awareness and avoided the need for training while calculating dataset similarity. Their method leveraged a combination of Euclidean distance for label distributions and Wasserstein distance for feature distributions to quantify the dissimilarity between data points They demonstrated correlation between dataset distance and transfer learning performance. However, their measure is unbounded and sensitive to the scale of the data points. Label-aware methods that require model training include methods that measure joint label-feature distributions (*e.g.,* JDOT Courty et al. (2017), DeepJDOT Damodaran et al. (2018)) and methods that measure dataset dynamics via gradient flow Alvarez-Melis & Fusi (2021). The requirement for model training makes these methods task specific. Further, these methods require access to the data points to estimate the underlying distributions, creating privacy concerns in domains like healthcare. Label unaware methods that do not require model training have used the Gromov-Wasserstein distance Mémoli (2011; 2017). This distance is well-suited for cross-domain dataset comparisons, but incorporating label information has been proven challenging. Pándy et al. (2021) developed a bounded but label-unaware method that requires training. This method uses embeddings of data points and calculates the Bhattacharyya distance within these embedding space. This builds upon previous works that demonstrated the benefits of embedding complex data points in Wasserstein space for improving similarity calculations using OT Frogner et al. (2019); Muzellec & Cuturi (2018). However, these approaches require accurate embedding of each class, making them sensitive to the chosen distribution space and embedding technique, while also introducing computational complexity.

Discrepancy measures and parameter sensitivity have also been used to quantify similarity in domain adaptation and meta-learning, respectively Cortes & Mohri (2011); Achille et al. (2019); Ben-David et al. (2006); Mansour et al. (2009); Achille et al. (2018); Liang et al. (2019); Li (2006). Discrepancy measures depend on learning a function that maps features from the source to the target domain. It works well in providing generalization bounds for adaptation. However, it requires careful selection of the hypothesis class and a large sample size to train an accurate mapping function. Parameter sensitivity based methods use information-theoretic measures to quantify model sensitivity to specific parameters. Recent work has combined parameter sensitivity with OT to create a coupled transfer distance for model weights Gao & Chaudhari (2021). These approaches require training of a specific probe network, making them specific to the learning task at hand.

## 4 BACKGROUND

### 4.1 OPTIMAL TRANSPORT

OT is a framework that quantifies the dissimilarity between probability distributions Villani et al. (2009). It achieves this by defining a cost function between distributions and then identifying the optimal transportation map that minimizes this cost. It has desirable theoretical and practical properties Genevay et al. (2018). In particular, it considers probability measures irrespective of their support. Further, the probability measures can be either continuous or discrete. The formulation for discrete probability measures is especially useful in settings with high dimensional data or finite samples. The Kantorovich (1960) formulation of OT between two distributions $X$ and $Y$ is shown in eq.S.1. In the discrete case, where the full joint distribution is unknown, the pairwise cost between data points is represented as an $n \times m$ matrix where $C_{i,j} = c(x_i, y_j)$. In this case, eq.S.1 has a cubic computational complexity and is replaced by an entropy-regularized Sinkhorn formulation that has quadratic complexity Cuturi (2013):

$$OT(X,Y) := \min_{\pi \in \Pi(X,Y)} \int_{\mathcal{X} \times \mathcal{Y}} c(x,y) d\pi(x,y) + \epsilon H(\pi | X \otimes Y) \tag{1}$$

where $H(\pi | X \otimes Y)$ is the relative entropy. Note that the parameter $\epsilon$ can be tuned to strike a balance between computational speed and accuracy. A higher $\epsilon$ decreases computation time but also reduces the ability to identify the optimal coupling.

## 4.2 HELLINGER DISTANCE

The Hellinger Distance (HD) is a measure of similarity between two probability distributions.HD is used in applications, such as domain adaptation, semi-supervised clustering, feature selection, and to quantify label similarity Chen et al. (2023); Goldberg et al. (2009); Yin et al. (2013); González-Castro et al. (2013). HD offers the advantage of robustness and insensitivity to skew, making it valuable in real-world data scenarios.Cieslak et al. (2012); Lindsay (1994). HD can be calculated as in eq.S.3. For two Gaussian distributions, HD has the closed form solution found in eq.2

## 4.3 SECURE MULTIPARTY COMPUTATION

Secure multiparty computation (SMPC) is a cryptographic technique that enables multiple parties to jointly compute a function without revealing their inputs to each other Evans et al. (2018)(see Section S.1.4). The advantage of SMPC is its ability to protect privacy against both outside adversaries and other involved parties, all while producing accurate outputs.

## 4.4 DIFFERENTIAL PRIVACY

Differential Privacy (DP) is a statistical technique that provides privacy by adding random but controlled noise to the data or its outputs Dwork et al. (2014). A mechanism satisfies DP if, for any pair of datasets that are identical except for one data point, the likelihood of producing a specific output from this mechanism varies by no more than a factor of $e^\epsilon$ (smaller $\epsilon$ indicates stronger privacy). zero-concentrated DP (zCDP) is a relaxation of pure DP that provides tighter analytic bounds on the privacy loss given certain assumptions are made on the distribution Bun & Steinke (2016); Dwork & Rothblum (2016) (see Section S.1.5 for more details).

## 5 PROPOSED FRAMEWORK

Our approach is summarized in algorithm 1. Overall, in supervised learning, our objective is to learn a function $f : \mathcal{X} \mapsto \mathcal{Y}$ where $\mathcal{X}$ is the feature space and $\mathcal{Y}$ is the label set. A dataset consists of feature-label pairs $(x, y)$, where $x \in \mathcal{X}$ and $y \in \mathcal{Y}$ (we use $z$ to denote the $(x, y)$ pair). In our analysis, we consider two distinct datasets, $D_1$ and $D_2$. For $D_1$ and $D_2$, the feature vectors are denoted as $x_i$ and $x_j$ and the corresponding labels are represented as $y_i^c$ and $y_j^{c'}$, respectively. Here $i$ and $j$ refer to indices of the specific data points and $c$ and $c'$ refer to label *classes* in $D_1$ and $D_2$ . Our goal is to calculate the OT score between two datasets using a cost function producing a bounded score to capture their similarity. To do this, we replace Wasserstein distance with Cosine Similarity-Hellinger distance as the OT cost function. We propose using Cosine similarity and Hellinger distance to capture feature vector and label distribution differences, respectively. Comparing datasets by measuring OT scores between feature-label pairs is first described by Alvarez-Melis & Fusi (2020).

There are several considerations and assumptions pertaining to the procedures *FEATURE COST CALCU-LATION*, *LABEL COST CALCULATION* and *OPTIMAL TRANSPORT CALCULATION*. In procedure *FEATURE COST CALCULATION*, when data points have dimension $\geq 1000$, we recommend using an autoencoder to learn compressed representations of the data. In most cases, this has the advantage of rendering the vectors linearly separable, thereby improving the ability of cosine similarity to capture accurate similarity between vectors. In procedure *LABEL COST CALCULATION*, we assume a Gaussian label distribution in our datasets, as done by Alvarez-Melis & Fusi (2020). This assumption is based on prior research and supported by the fact that the Gaussian approximation serves as an upper bound for the true cost and is effective for a wide range of labels Gelbrich (1990); Seddik et al. (2020). Note that the boundedness of the score remains unaffected by the label distribution. In *OPTIMAL TRANSPORT CALCULATION*, our approach has two hyperparameters, $\epsilon$, the regularization parameter for the Sinkhorn algorithm; and the normalization constant, $\lambda$. We recommend using a sufficiently low $\epsilon < 1e - 2$ to ensure the accuracy of the optimal transport mapping (see Section S.5.5.1). For $\lambda$, we set it to 3 as the cosine similarity and Hellinger distance range from 0-2 and 0-1, respectively. However, we show that our dataset similarity score is robust to a wide range of feature-to-label cost ratios, ranging from 1:4 to 4:1 (see Section S.5.5.2).

### 5.1 PRIVACY-PRESERVING CALCULATION IN FEDERATED SETTINGS

To overcome data-sharing limitations when calculating similarity between datasets in federated settings, we implement SMPC to calculate feature costs and zCDP to calculate label costs. Our SMPC method, adapted from Du et al. (2004), efficiently calculates feature costs through joint dot product calculations. Our zCDP approach, introduced by Biswas et al. (2020), ensures accurate label cost computation with low sample complexity, making it ideal for multi-source learning scenarios. Our primary contribution lies in the development of a cost metric

---

**Algorithm 1** Computing Optimal Transport Scores between Datasets $D_1$ and $D_2$

---

1: **Input:** Datasets $D_1$ and $D_2$, regularization parameter $\epsilon$, normalization parameter $\lambda$
2: **Output:** Overall dataset similarity score
3: **procedure** FEATURE COST CALCULATION
4:     **for** each data point $x_i$ in $D_1$ and $x_j$ in $D_2$ **do**
5:         **if** data dimensionality $\geq 1000$ **then**
6:             Embed data into smaller dimension (see SectionS.2.1)
7:         **end if**
8:         **if** datasets should remain private **then**
9:             Calculate the cosine similarity (see eq.S.2) using SMPC (see SectionS.2.2)
10:         **else**
11:             Calculate the cosine similarity (see eq.S.2)
12:         **end if**
13:     **end for**
14: **end procedure**
15: **procedure** LABEL COST CALCULATION
16:     **for** each label class pair $y^c$ in $D_1$ and $y^{c'}$ in $D_2$ **do**
17:         Estimate the distribution parameters: Mean, $\mu$, and Covariance, $\Sigma$ (see Section S.2.4)
18:         **if** datasets should remain private **then**
19:             Add zCDP noise to parameters (see Biswas et al. (2020) and Section S.2.3)
20:         **end if**
21:         Compute the Hellinger distance

$$H^2(y^c, y^{c'}) = 1 - \frac{|\Sigma^c|^{\frac{1}{4}}|\Sigma^{c'}|^{\frac{1}{4}}}{|\frac{1}{2}\Sigma^c + \frac{1}{2}\Sigma^{c'}|^{\frac{1}{4}}} exp[-\frac{1}{8}\mu^\top(\frac{1}{2}\Sigma^c + \frac{1}{2}\Sigma^{c'})^{-1}\mu] \tag{2}$$

22:     **end for**
23: **end procedure**
24: **procedure** CREATION OF TOTAL COST MATRIX
25:     **for** each pair of feature-label points **do**
26:         Compute the total feature-label cost

$$c(z_i, z_j) = Cos_{sim}(x_i, x_j) + H^2(y_i^c, y_j^{c'}) \tag{3}$$

27:     **end for**
28: **end procedure**
29: **procedure** OPTIMAL TRANSPORT CALCULATION
30:     Employ the regularized Sinkhorn algorithm to create a transportation map as in eq.1
31:     Calculate the overall similarity score

$$OT(D_1, D_2) = \frac{1}{\lambda} \sum_{z_i \in D_1, z_j \in D_2} c(z_i, z_j)\pi(z_i, z_j) \tag{4}$$

    where $\pi$ is the transportation map, $c(.,.)$ is the pairwise cost, and $\lambda$ is the normalization constant.
32: **end procedure**

---

that can be made privacy-preserving with minimal computational overhead (see Section S.5.4.5 and Biswas et al. (2020)). Further, we show these method achieve high accuracy with low sample complexity (see Fig. S.1, Fig. S.2). For a full treatment of the privacy and security guarantees and complexity please refer to the original papers Du et al. (2004); Biswas et al. (2020). The incorporation of privacy-preservation is particularly important for domains where MSL has become essential, *e.g.,* healthcare Dayan et al. (2021); Rieke et al. (2020).

## 6 THEORETICAL INSIGHTS

### 6.1 UNDERSTANDING COSINE SIMILARITY AS AN APPROXIMATION FOR DATASET SIMILARITY IN DEEP LEARNING

Cosine similarity is a commonly used metric in machine learning, especially in high dimensional settings as it is invariant to scaling effects Strehl et al. (2000). We show that in high dimensional spaces, cosine similarity serves as a valuable indicator of shared information among data points in different datasets. This is

because the boundedness of orthogonality for random vectors is remarkably tight at high dimensions, due to the 'concentration of measure' phenomenon Ledoux (2001). We expand this bound from two random vectors to encompass any pair of vectors drawn from two random and independent datasets (see Section S.3.1). Specifically, when two datasets are independent and lack shared features, the probability that a randomly selected pair of vectors from these datasets exceeds a cosine threshold of $t$ is as follows:

$$Pr\left(x_i \cdot x_j = |cos\theta_{x_i,x_j}| > t\right) < \frac{Var[x_i \cdot x_j]^{\frac{1}{n}}}{t^2} \approx \frac{1}{nt^2} \tag{5}$$

where $x_i \in D_i, x_j \in D_j$ with feature space size $n$. This bound requires high dimensional vector spaces (*e.g.* $e^{0.01n}$). Crucially, the validity of this bound hinges on the assumption that the datasets being compared have independent means and covariances. In scenarios where two datasets are related, this assumption may not hold and vectors may be non-orthogonal in expectation. It is precisely this violation of orthogonality that forms the basis for our choice of cosine similarity as a metric to assess dataset similarity.

Next, we demonstrate how models leverage the non-orthogonality of related data points to align their weights during training. Prior research has already established that neural networks make use of measures akin to cosine similarity for learning Liu et al. (2017); Luo et al. (2018). This can be attributed to the widespread use of activation functions that employ the dot product between feature vector and weight, $\phi(w \cdot x_i)$, where $\phi$ denotes the activation function. Since both weights and features are often normalized for stability, this dot product approximates the cosine of the angle between the vectors. In this work, we show that the alignment of model weights relies on the relative orientation of data points due to this dot product. In particular, the learning process via gradient descent is defined as $w_{t+1} = -\eta\nabla_w L(w_t, x_i, y_i)$ where $L(.,.,.)$ is the loss and $\nabla_w$ represents the model gradients. By breaking down each data point into a parallel and perpendicular component relative to the weight vector, $w$, we find that the change in model weights can be described as $\Delta w \propto x_{\parallel}$. Without loss of generality, when considering two feature vectors, $x_1$ and $x_2$, from the training set, the combined change in $w$ can be described as $\Delta w \propto \sum_{i=1,2} x_{i\parallel}$. Assuming normalized vectors, as described above, this is $\propto cos(\theta_{x_1,w} + \theta_{x_2,w}) \propto cos_{sim}(x_1, x_2)$. When the cosine of the angle between $x_1$ and $x_2$ is close to zero, it implies a negligible change in $w$, leading to minimal improvement in loss (for a comprehensive discussion, including the impact of non-linearities introduced by activation functions and deep networks, please refer to Section S.3.2).

To gain further insight into this relationship, we can examine how model gradient diversity changes when models are trained on different datasets. Given that gradients closely reflect the learning task, correlation between gradient diversity and our metric serves as an indicator that our metric accurately identifies task similarity Simonyan et al. (2013); Wang et al. (2020). Importantly, our metric achieves this without model training. The concept of gradient diversity is introduced by Yin et al., Yin et al. (2018) (see eq.S.14. High gradient diversity is indicative of models learning in divergent directions, due to dataset or task dissimilarity Zhang et al. (2021); Wang et al. (2020). For empirical results, please refer to Section 7.4.

### 6.2 MOTIVATING THE USE OF HELLINGER DISTANCE

We use HD to assess label similarity for three main reasons: it is bounded, it aligns with the concept of Optimal Transport, and it provides a desirable interpretation of label similarity. The standard Wasserstein distance is a special case of the $(f, \Gamma)$-divergence family, where the $\Gamma$ cost function is bounded 1-Lipschitz functions, such as the L2 norm Birrell et al. (2022). However, the $(f, \Gamma)$-divergence framework allows the use of alternative admissible $\Gamma$ functions. Given we use the HD to generate the cost map, and HD uses a bounded, continuous function, it can be substituted. The value of employing HD lies in its ability to shift focus from distribution differences using Euclidean geometry, to a comparison of probability densities within the statistical manifold. This is because HD is intrinsically connected to the Riemannian metric defined on the statistical manifold of probability distributions and effectively captures the shortest geodesic distance in this manifold Baktashmotlagh et al. (2014). Incorporating the Hellinger distance into our analysis allows us to compare label distributions in a manner that retains the valuable aspects of the Wasserstein metric, such as its interpretation in terms of optimal transport, while ensuring that our metric remains bounded and relevant in the context of probability spaces.

## 7 EXPERIMENTS

### 7.1 SETUP: DATASETS, TRAINING, AND MODELS

We link our dataset similarity score to MSL performance across seven datasets of different dimensions and scale. We include three tabular datasets and four imaging datasets: synthetic, credit fraud, weather, EMNIST, CIFAR, IXITiny and ISIC2019 Worldline & the Machine Learning Group of Université Libre de Bruxelles; Malinin et al. (2021); Cohen et al. (2017); Krizhevsky et al. (2009); Brain-development.org; Tschandl et al. (2018); Combalia et al. (2019); Codella et al. (2018). The synthetic dataset allows us to precisely control distribution shifts and compare scores across a range of scenarios, while the latter are benchmark datasets used to evaluate model performance in non-IID data scenarios. Below we provide a brief overview of the datasets, models, and tasks

(Table 1. Please see S.4.1 for full details. All code for data partitioning, metric calculation and model training can be found at `github.com/iclr-otcost-submission/submit`.

| Dataset | Description | Dimension | Model | Outcome | Metric |
|---------|-------------|-----------|-------|---------|--------|
| Synthetic | - | 12 | MLP | binary | F1-score |
| Credit | Fraud prediction | 28 | MLP | binary | F1-score |
| Weather | Temperature prediction | 123 | MLP | continuous | $R^2$ |
| EMNIST | Handwritten digits & characters | 28x28 | LeNet5 | multi-class | Accuracy |
| CIFAR | Colour images | 1x32x32 | ResNet-18 | multi-class | Accuracy |
| IXITiny | 3D-brain MRIs | 1x83x44x55 | 3D-Unet | image segmentation | DICE |
| ISIC2019 | Dermoscopy images | 3x224x336 | EfficientNet | multi-class | Balanced Accuracy |

Table 1: Summary of datasets, models, and outcome metrics used in our study.

### 7.1.1 SETUP: DATASETS

Several considerations underpin our dataset creation approach:

- **Data partitioning**: For synthetic, credit, weather, EMNIST and CIFAR datasets, we create multi-source datasets with varying levels of distribution shift by partitioning the original dataset and varying label distributions within each dataset. For synthetic and credit datasets, we also introduce feature skew by incorporating Gaussian noise, following the approach outlined in Zhao et al. (2018) (see Section S.4.1.1). The IXITiny and ISIC2019 datasets naturally exhibit non-IID partitions based on the site of data collection. For these datasets, we adhere to the data pre-processing and modelling procedures outlined in Terrail et al. (2022).

- **Dataset compression**: For CIFAR, IXITiny and ISIC2019, we train autoencoders to obtain compressed representations of the data with dimensions of 512, 1,024, and 18,432. These representations are used to calculate dataset similarity scores. Specifically for CIFAR and IXITiny, we train new autoencoder models. For ISIC2019, we use a pre-trained model using a VGG-19 model backbone Zenn (2021) (see Section S.2.1).

### 7.1.2 SETUP: TRAINING AND MODELS

For each of the previously mentioned tasks, we asses the performance using five different training approaches: (1) Single training: This serves as our baseline, where a model is trained on a single dataset. (2) Joint training: One model is trained on the combined dataset from both sites. (3) Federated learning using Federated Averaging (FedAvg): Two models are trained and aggregated as described by McMahan et al. (2017). (4) Personalized Federated Learning with Moreau Envelopes (pfedme): A meta-learning based FL method that decouples the local and global models using Moreau envelopesT Dinh et al. (2020). (5) Ditto Li et al. (2021): Another personalised FL variant that trains separate local and global models simultaneously while encouraging the local model to stay close to the global model. The pfedme and Ditto algorithms allow us to explore whether non-IID learning algorithms can mitigate the impact of dataset dissimilarity on model performance.

For each combination of dataset, dataset dissimilarity score, and training, we performed hyperparameter tuning through grid search. This involved exploring five learning rates ($5e^{-1}, 1e^{-1}, 5e^{-2}, 1e^{-2}, 5e^{-3}, 5e^{-4}$) and two optimizers: ADAM and SGD (see Section S.4.1.2). Note that the SGD optimizer is exclusively used for federated algorithms as it can improve convergence in some cases. For pfedme and Ditto, we also conduct this grid search over the regularization parameters: $1e^{-3}, 1e^{-2}, 1e^{-1}, 5e^{-1}, 1$. This parameter controls the level of personalization *vs.* aggregation during training. Our training consists of a maximum of 500 epochs, with early stopping implemented if there is no improvement in validation loss after 5 epochs. For the synthetic, credit, and weather datasets, we conducted 500 experiments, while for CIFAR and EMNIST, we performed 50 experiments. For these datasets, we present the median results, accompanied by 95% confidence intervals. As for IXITiny and ISIC2019, we ran 3 experiments and present the median results.

### 7.2 RESULTS: OUR DATASET SIMILARITY SCORE CAPTURES REAL-WORLD DIFFERENCES IN DATASETS

For IXITiny and ISIC2019 where the datasets naturally partition into non-IID datasets based on real-world data collection processes, it is useful to compare our OT based similarity score with these data collection practices.

- For IXITiny, we find that the scores produced by our metric align with real-world data collection practices. Specifically, two sites have MRI images collected on a similar machine (Philips) with similar imaging parameters. In contrast, the third site has images collected using a different machine (GE) with unspecified imaging parameters Brain-development.org. Our metric assigns a score of 0.08 between sites 1 and 2, and 0.28 and 0.30 between sites 3 and 1 and sites 3 and 2, respectively. This is consistent with the observed image intensity distributions across the three sites Terrail et al. (2022).

- For ISIC2019, we find that our score aligns with our understanding of how the data was collected while also providing some novel insights. ISIC2019 data is collected from four sites (two in Europe, one in USA, one in Australia). One of the European sites uses three different machines, resulting in separate data partitions. We find that European sites have lower scores with each other than with Australian and American sites, consistent with the findings in Terrail et al. (2022). Additionally, we identify that imaging machine has a significant impact on dataset similarity. Specifically, datasets from the same European site collected using different imaging machines are more dissimilar than datasets from different European sites collected using a similar machine Wen et al. (2022). Our empirical MSL model performance results corroborate this finding. This highlights the need to consider not only the data source but also the specific devices used for data collection.

## 7.3 RESULTS: OUR DATASET SIMILARITY SCORE CORRELATES WITH MODEL PERFORMANCE

Figure 1 shows the performance of models trained on datasets with varying similarity scores. We observed that the scores for most dataset partitions fall between 0 to 0.5. This is expected as a score of 0.5 implies no similarity between vectors and total randomness in their shared information. Scores between 0.5 and 1 are achievable using the synthetic datasets and are associated with extreme negative learning (see Fig.S.5). Our findings indicate that scores ranging between 0 to 0.2 generally result in improved performance in MSL, while scores exceeding 0.3 lead to a decline in model performance compared to the baseline. This finding is consistent across all datasets tested and different learning algorithms, indicating the robustness of our similarity metric. We demonstrate that while non-IID algorithms such as pFedMe and Ditto can offset the negative effects of high cost between datasets, they might reduce the benefits of collaborative training in scenarios with lower cost. Notably, our metric exhibits a correlation with the regularization parameter: datasets with higher similarity (low cost) tend to have larger regularization parameters, promoting aggregation, whereas more dissimilar datasets (high cost) favor lower regularization values, leaning towards personalization (see Section S.4.1.2). This finding underscores the importance of incorporating a prior assessment of dataset similarity into loss-based non-IID algorithms. Note, while model-based approaches (*e.g.* Meta-learning) implicitly assess dataset similarity during training, these approaches often require estimation of complexity measures for a function class (*e.g.* Rademacher complexity), which requires larger sample sizes for accurate estimation. This poses a challenge in MSL when individual sites' sample sizes are small, as such we do not assess model-based approaches here.

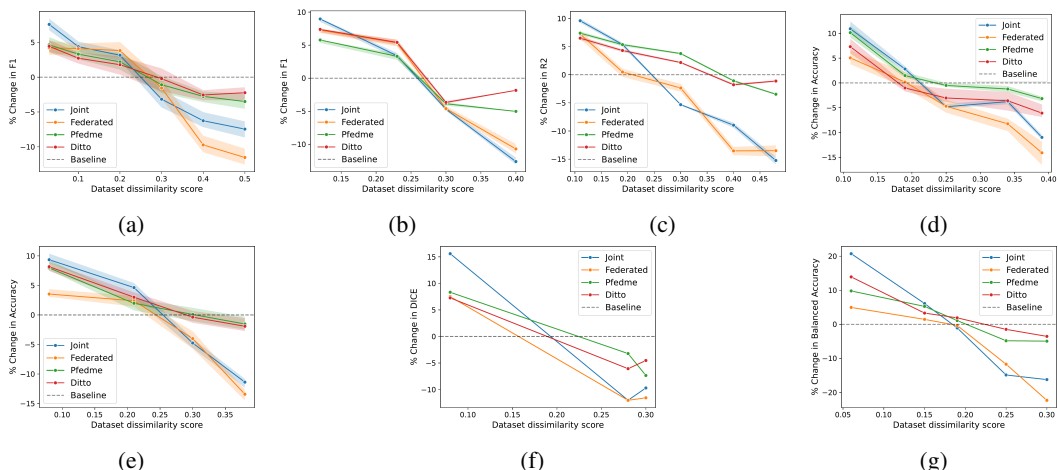

Figure 1: **Performance of models trained on datasets with different scores**. Results expressed as percentage changes from single baseline with Joint (blue), Federated (orange), pfedme (green) and Ditto (Red). Results shown for synthetic (a), credit (b), weather (c), EMNIST (d), CIFAR (e), IXITiny (f), and ISIC2019 (g).

## 7.4 RESULTS: OUR DATASET SIMILARITY SCORE CORRELATES WITH MODEL GRADIENT DIVERSITY

In order to gain a deeper understanding of the relationship between model performance and our metric, we examine the changes in model gradient diversity for federated models trained on datasets with differing similarity

scores. Our findings demonstrate a correlation between gradient divergence and our metric. Datasets with higher dataset dissimilarity result in greater divergence, and this divergence manifests earlier (Figure 2). Further, we find that our metric has higher correlation with gradient diversity than standard Wasserstein distance across all datasets. This result is statistically significant (paired t-test, $p = 0.028$). These results suggest that our metric for dataset similarity captures training dynamics, thereby establishing a connection to why it correlates with model performance, even without the need for model training.

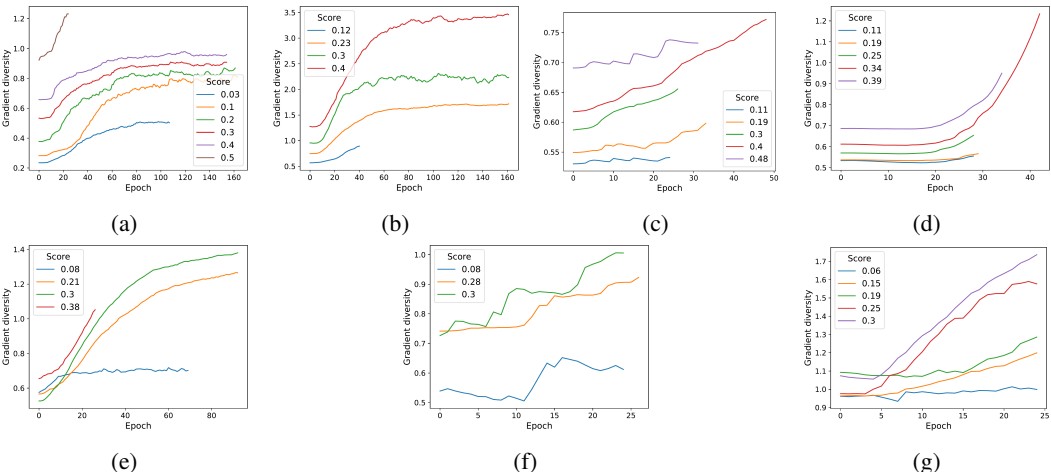

Figure 2: **Model gradient diversity during training for Federated (FedAvg) models**. Results shown for synthetic (a), credit (b), weather (c), EMNIST (d), CIFAR (e), IXITiny (f), and ISIC2019 (g).

## 7.5 RESULTS: COMPUTATIONAL AND SAMPLE SIZE COMPLEXITY

The computational complexity of calculating our metric is determined by the Sinkhorn algorithm and is $O(n^2/\epsilon^2)$, where $n$ is the number of samples in a dataset and $\epsilon$ is the Sinkhorn regularization parameter. In practical terms, the runtime is most sensitive to $\epsilon$ as the Sinkhorn algorithm is sample size efficient and can be estimated on a subsample of the dataset (see Mena & Niles-Weed (2019); Genevay et al. (2019) for theoretical bounds). Fig.S.3 presents a comparison of the estimated scores on the full datasets vs. subsampled datasets. We show that even for high dimensional datasets, accurate estimates can be achieved with approximately 1,000 samples. Regarding the choice of $\epsilon$, we set it to $1e^{-3}$ in our our experiments. While a large $\epsilon$ reduces runtime, it also leads to a less optimal transport map. It has been shown theoretically that as $\epsilon \to \infty$, the transport map between datasets becomes uniform Peyré et al. (2017). We empirically show this in Fig.S.7. Throughout our experiments, we are able to calculate our metric within ~30 minutes on 1 Intel Xeon CPU E5-2699.

## 7.6 RESULTS: ADVANTAGES OF OUR MEASURE OVER TRADITIONAL MEASURES

While alternative metrics such as Wasserstein distance and KL-divergence exist, the advantage of our metric lies in its consistent interpretation across datasets of varying dimension and scale. To evaluate this, we compare our metric with Wasserstein distance and KL-divergence. Although both of these metrics also show a relationship between increasing scores and performance, the key distinction is that the scores they produce are not consistently interpretable (see S.5.4). The same score can imply either improved or deteriorated model performance, depending on the dataset. For example, a Wasserstein cost of $\sim 1e^8$ results in improved learning in IXITiny but negative learning in ISIC2019. This variability makes it challenging for practitioners to interpret the score. Further, our metric can be computed with low computational overhead in a privacy-preserving manner using SMPC. In comparison, calculating Euclidean distance using SMPC would entail higher computational costs (see Section S.5.4.5), and as of now, no SMPC methods have been developed for KL-divergence.

**Limitations:** There are several limitations to our current approach that can be explored in future research. First, our metric assumes concordant features across the datasets. Future work could explore how our metric performs with datasets with different features. It is worth noting, however, that in practical MSL scenarios, datasets typically have identical structures, with distribution shifts primarily attributed to other factors. Second, the computational complexity of our metric increases with the dataset dimensionality. Third, the use of cosine similarity limits our ability to capture non-linear relationships within the features. The latter two limitations can be addressed by embedding the datasets to reduce dimensionality and ensuring linear separability of vectors prior to metric computation. Our successful results with CIFAR, IXITiny and ISIC2019 datasets support the viability of this approach. Finally, we modeled all label distributions as Gaussian. While this assumption held well our datasets, it may not be suitable for all scenarios.

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
