# SUPPLEMENT

## S.1 BACKGROUND

### S.1.1 OPTIMAL TRANSPORT

The Kantorovich (1960) formulation of OT between two distributions $X$ and $Y$ is:

$$OT(X,Y) := \min_{\pi \in \Pi(X,Y)} \int_{\mathcal{X} \times \mathcal{Y}} c(x,y) d\pi(x,y) \tag{S.1}$$

Where $c(x,y)$ is the cost function between two points and $\Pi(X,Y)$ is the couplings over the product space $\mathcal{X} \times \mathcal{Y}$ with marginals $p(x)$ and $p(y)$. In euclidean metric space, $c(x,y)$ can be defined using euclidean distance, *i.e.,* $|x-y|_p$. In this case, OT cost is known as the p-Wasserstein distance.

### S.1.2 COSINE SIMILARITY

Cosine similarity is a widely used metric in machine learning to quantify the similarity between two vectors. Mikolov et al. (2013). Cosine similarity between two vectors $x_i$ and $x_j$ is calculated as:

$$\frac{x_i \cdot x_j}{||x_i|| ||x_j||} \tag{S.2}$$

The value ranges from -1 to 1 with 0 meaning the vectors are orthogonal. To ensure non-negativity of the score, it is common to instead use 1- cosine similarity which ranges from 0-2.

### S.1.3 HELLINGER DISTANCE

Given two probability distributions, $P$ and $Q$, defined over the same domain $\mathcal{X}$, HD can be calculated as:

$$H^2(P,Q) = \frac{1}{2} \int_{\mathcal{X}} \left( \sqrt{P(dx)} - \sqrt{Q(dx)} \right)^2 \tag{S.3}$$

### S.1.4 SECURE MULTIPARTY COMPUTATION

Consider three individuals, Alice ($p_1$), Bob ($p_2$), and Carol ($p_3$), who wish to calculate the mean of their ages while preserving the privacy of their individual ages. To achieve this, each party independently splits their age into three random values, denoted as $x_1$, $x_2$, and $x_3$, such that their age, $x$, is the sum of these values ($x = x_1 + x_2 + x_3$). Each party then shares two of their secret values with the other parties, ensuring that each party holds one secret share from every individual. This is coordinated such that $p_i$ holds $x_i$ from every party. Next, each party computes the mean of their secret shares. Finally, all parties share their computed values, which are summed to reveal the mean age without disclosing any individual ages. Note, this is an illustrative example. SMPC uses additional cryptographic techniques on the secret shares such as encryption, secure function evaluation, and secure communication to preserve privacy.

### S.1.5 DIFFERENTIAL PRIVACY

Differential privacy is a formal standard for protecting individual privacy in statistical data analysis. It has two forms pure and approximate. Pure DP is parameterized by one parameter, $\epsilon$, that quantifies the allowable privacy loss a sample may experience Dwork et al. (2014). The formulation for pure DP is:

$$\frac{Pr[F(x) = S]}{Pr[F(x') = S]} \le e^\epsilon \tag{S.4}$$

Where $x$ and $x'$ are neighbouring datasets differing by at most one sample, $F(.)$ is a function applied to the dataset, and $S$ is the output. $e^\epsilon$ is the multiplicative factor where a lower $\epsilon$ represents greater privacy guarantees. Approximate or $(\epsilon, \delta)$-DP is a more flexible variant that guarantees that the probability of exceeding the privacy loss bound specified by $\epsilon$ is at most $\delta$ Dwork et al. (2006). The formulation for approximate-DP is:

$$Pr[F(x) = S] \le e^\epsilon Pr[F(x') = S] + \delta \tag{S.5}$$

When $\delta = 0$, pure and approximate-DP are equivalent.

zero Concentrated DP (zCDP) is an extension of approximate-DP that permits sharper analysis when the mechanism is Gaussian with small mean Bun & Steinke (2016); **?**. A randomized mechanism satisfies zCDP if for all neighboring datasets and all $\alpha \in (1, \infty)$:

$$D_\alpha(F(x)||F(x')) = \frac{1}{\alpha - 1} log E(\frac{Pr[F(x) = S]}{Pr[F(x') = S]})^\alpha \le \rho\alpha \tag{S.6}$$

Here, the first term is the Rēnyi divergence, a generalized measure of divergence between probability distributions, $\rho$ is the privacy parameter, and $\alpha$ is a tuning parameter that adjusts the sensitivity of the privacy guarantee. To convert between $\rho$ and $\epsilon, \delta$ one can use the following formula Near & Abuah (2021):

$$\epsilon = \rho + 2\sqrt{\rho\log(\frac{1}{\delta})} \tag{S.7}$$

## S.2 ALGORITHM

### S.2.1 EMBEDDING PROCESS

When working with high-dimensional data, especially with vectors of dimensions $\geq 1000$, embedding can provide a more compact and efficient representation. In our experiments, this threshold has proven effective, although it remains a heuristic recommendation. The added benefit of embedding vectors is that, for features that are highly non-linear and multi-dimensional *e.g.,* images, embedding compresses the information into a single dimension vector. We apply this approach for three datasets. All training details can be found at `github.com/iclr-otcost-submission/submit`

- **CIFAR**: We train an autoencoder with 7 layers in the encoder block and 7 layers in the decoder block. We tested bottleneck sizes of 250, 500, 750, 1,000, and 2,000. We selected 1,000 as it produced the lowest reconstruction error.

- **IXITiny**: We train an autoencoder with 4 layers in the encoder block and 3 layers in the decoder block. Rather than recreating the image, we train an autoencoder to recreate the bottleneck representation from a pretrained 3D Unet that is used for brain MRI segmentation **?**. We do this as we assume the 3D Unet has learned useful feature representations. We tested bottleneck sizes of 256, 512, 1,024, 2,048 and 4,960. We selected 2,048 as it produced the lowest reconstruction error.

- **ISIC2019**: We use a pre-trained autoencoder with a VGG-19 backbone from Zenn (2021). We found this to perform better than training our own model. We use the bottleneck representation from the pre-trained model which produces vectors with an embedding size of 18,432. This allows us to test the accuracy of our metric on very high dimensional vectors.

### S.2.2 SMPC PROTOCOL

Our SMPC protocol to calculate the dot product in a privacy preserving way is adapted from Du et al. (2004).

---

$Site_1$ hold dataset A ($N_1 \times d$), $Site_2$ hold dataset B ($N_2 \times d$) where $d$ =feature dimension and $N_i$ =number of patients.

1. Server creates a random invertible matrix $M_{d \times d}$ using Reed-Hoffman encoding and sends $M$ to Site 1 and $M^{-1}$ to Site 2.

2. Site 1 computes $A_1 = A \times M_{\text{left}}$, $A_2 = A \times M_{\text{right}}$ and sends $A_1$ to the server.

3. Site 2 computes $B_1 = B \times M_{\text{top}}^{-1}$, $B_2 = B \times M_{\text{bottom}}^{-1}$ and sends $B_2$ to the server.

4. Server sends $B_2$ to Site 1 and $A_1$ to Site 2.

5. Site 1 computes $V_a = A_2 \times B_2$ and sends it to the server.

6. Site 2 computes $V_b = A_1 \times B_1$ and sends it to the server.

---

We have $\mathbf{AB} = AMM^{-1}B = \begin{pmatrix} A_1 & A_2 \end{pmatrix} \begin{pmatrix} B_1 \\ B_2 \end{pmatrix} = V_a + V_b$.

The method requires construction of a secure matrix $M$ which can be generated using maximum distance separable (MDS) codes such as Reed-Solomon codes. MDS codes ensure that any subset of columns are linearly independent of each other such that it is impossible to recover the original data MacWilliams (1977).

### S.2.3 zCDP PROTOCOL

We use an iterative method called COINPRESS developed by Biswas et al. (2020), which enables release of the mean and covariance matrices of multivariate data. Note, while the method works best for sub-Gaussian data, it holds for other distributions with fatter tails. For a technical explanation of the method including privacy guarantees please see Biswas et al. (2020). The method requires the user to input *a priori* knowledge of the data including starting values and a search radius. We use the actual label class mean and covariance for starting values and a search radius of 1. This is possible as, in the case of estimating label similarity, our 'population' is

the dataset, thus we confidently know the true population value, *i.e.,* we have a strong prior on the summary statistics. This helps to reduce the number of iterations necessary and improves the accuracy of the outputted values. Subsequently, we use a $\rho = 0.1$ which translates to $\epsilon = 0.99, \delta = 0.01$.

### S.2.4 LABEL COST CALCULATION

The label distance calculation in our metric and in Alvarez-Melis & Fusi (2020) requires the inversion of covariance matrices. For high-dimensional and sparse datasets with singular covariance matrices, this can result in degenerate matrices. To avoid this, we use PCA to reduce the dimensionality of the covariance matrices while retaining 80% of the explained variance. This allows us to calculate the label scores using the majority of the information while ensuring covariance matrices are non-singular. Importantly, for privacy preservation, we fit the principal components on one covariance matrix and apply the same transformation to the other, eliminating the need to share raw data. If both covariance matrices need fitting together, we suggest using Federated PCA Grammenos et al. (2020).

## S.3 THEORETICAL INSIGHTS

### S.3.1 BOUND OF ORTHOGONALITY

#### S.3.1.1 PRELIMINARIES

The proof relies on the following preliminaries:

1. A random unit vector $x = (x_1, x_2, ...x_n)$ on the the unit hypersphere of $\mathbb{R}^n$ can be approximated by randomly choosing each coordinate, $x_k$ from the set $\{1, -1\}$ for $\forall k \in 1, ..., n$ and then scaling by $\frac{1}{\sqrt{n}}$. Taking all coordinates $x_k \in \{1, -1\}$ is equivalent to assuming each coordinate, $k$, has a shifted and scaled Bernoulli distribution with $p = 0.5$. Note that we relax this assumption later in the proof. *Remark*: An exact way to select a random point on the surface is to choose $x_k$ from $\sim N(0, 1)$ and normalizing by $\frac{1}{(\Sigma_k x_k^2)^{0.5}}$. In high dimensions, the simpler Bernoulli method approximates this well.

2. A random unit vector inside the unit ball of $\mathbb{R}^n$ will almost surely be on the surface as the dimension of the unit ball, $n$, increases:

   - We define a unit ball $B_n := \{(x_1...x_n) : \sum x_i^2 \leq 1\}$ has volume $v(B_n) = \frac{\pi^{\frac{n}{2}}}{\Gamma(1+\frac{n}{2})}$. As $n \to \infty, v(B_n) \to 0$, as the gamma function in the denominator dominates the term.

   - The majority of the volume in $B_n$ is concentrated in a thin strip $S_n$ of width $O(\sqrt{\frac{c log n}{n}})$ around its boundary. Specfically, the volume in $B_n$ outside of this strip is approximately $1 - \frac{1}{c}$ for any $c > 1$ where $c$ is a scaling factor. This phenomenon is a manifestation of the "concentration of measure" phenomenon in high dimensions.

3. Consider two random vectors on the unit hypersphere in $\mathbb{R}^n$, denoted as $x_i$ and $x_j$. The dot product of these vectors is $x_i \cdot x_j = |x_i||x_j|cos(\theta)$ where $\theta$ is the angle between them. As $x_i$ and $x_j$ are both on the unit hypersphere, $|x_i| = |x_j| = 1$, so $x_i \cdot x_j = cos(\theta)$.

#### S.3.1.2 LEMMA 1: A FIXED VECTOR AND ANY RANDOM VECTOR ON THE SURFACE ARE ALMOST-ORTHOGONAL

Suppose $a$ is a fixed unit vector in $\mathbb{R}^n$ . Let $x$ be a random vector with $x_k \in \{1, -1\}$ such that $x = \frac{1}{\sqrt{n}}(x_1, ..., x_n)$. Then, let $X = a \cdot x = \sum_{k=1}^n a_k x_k$ such that $Pr(|X| > t) = Pr(Cos\theta_{a,x} > t)$. Then $E[X] = E[\sum a_i x_i] = 0$ and $Var[X] = E[(\sum a_i x_i)^2] - 0 = \sum a_i^2 E[x_i^2] = \sum \frac{a_i^2}{n} = \frac{1}{n}$. Assuming $X$ is normally distributed, the Chebyshev inequality of $X$ is:

$$Pr(|X| > t) < \frac{\frac{1}{n}}{t^2} = \frac{1}{nt^2} \tag{S.8}$$

as $n \to \infty, Pr(|X| > t) \to 0$. To use the Chebyshev inequality, we invoke the central-limit theorem, applicable when $n \to \infty$.

#### S.3.1.3 LEMMA 2: ANY 2 RANDOM VECTORS ON THE UNIT HYPERSPHERE ARE ALMOST ALWAYS ORTHOGONAL BASED ON CONCENTRATION OF MEASURE

Assume $x_i$ and $x_j$ are unit vectors drawn from a uniform distribution in the unit hypersphere. If we take $X_{i,j} = x_i \cdot x_j$ to be a point that lies in the ball. Given the concentration of measure phenomenon mentioned

in the preliminaries, the absolute value of $X_{i,j}$ is most likely approximately 0. Specifically, with probability greater than $1 - \frac{1}{n}$, the absolute value of $X_{i,j}$ is bounded by $O(\sqrt{\frac{\log n}{n}})$. The fraction of the volume of the unit ball lying outside the strip around the surface, where the absolute value of $cos\theta_{x_i,x_j}$ exceeds $\sqrt{\frac{\log n}{n}}$, is less than $\frac{1}{n}$. Thus, we have:

$$Pr\left(|cos\theta_{x_i,x_j}| > \sqrt{\frac{\log n}{n}}\right) < \frac{1}{n} \tag{S.9}$$

As $n \to \infty$ there is high probability every pair of vectors has inner product $\approx 0$.

### S.3.1.4  PROOF 1: ANY PAIR OF VECTORS SAMPLED FROM 2 RANDOM DATASETS ARE ALMOST ALWAYS ORTHOGONAL

While Lemma 2 gives us intuition into the concentration of measure phenomenon, it is limited to random vectors uniformly sampled from the unit hypersphere. To extend the proof to vectors sampled from 2 random datasets we can no longer assume the vectors are uniformly drawn from the unit hypersphere. Instead we assume that $D_1 \in \mathbb{R}^{n_1 \times f}$ and $D_2 \in \mathbb{R}^{n_2 \times f}$ where $n$ is the number of samples and $f$ is the number of features. We assume that each dataset is drawn from a distribution that is independent of the other, *i.e.,* the 2 datasets share no information and that both datasets have been standardized (a common practice in deep learning tasks). If we take $x_i$ and $x_j$ as vectors that come from datasets $D_1$ and $D_2$ such that $x_i \sim N(0, \Sigma_1)$ and $x_j \sim N(0, \Sigma_2)$. As $D_1$ and $D_2$ are independent to each other, the covariance matrices are also independent. Note that $\Sigma_1$ and $\Sigma_2$ are not necessarily diagonal matrices, implying that the components of the vectors in each dataset are not always independent. Using Lemma 1 and the fact the two vectors are independent of each other,

$E[x_i \cdot x_j] = E[x_i]E[x_j] = 0$ and $Var[x_i \cdot x_j] = E[(x_i \cdot x_j)^2] - 0 = E[\left(\sum_{k=1}^{f} x_{ik}x_{jk}\right)^2] = E[\Sigma_1\Sigma_2] \approx 0$.

The latter holds as the expectation of every entry $E[\Sigma_{1ik}\Sigma_{2kj}] = E[(x_{ik}x_{jk})^2] \approx 0$ when the datasets, and thus features, are completely independent and features are symmetrically distributed about 0. The latter assumption holds as most features are Z-score normalized. Then, using the Chebyshev inequality and the central limit theorem (applicable as $f, n >> 30$), we can bound the probability that the absolute value of the cosine of the angle between a pair of randomly drawn vectors from the two datasets exceeds a threshold $t$:

$$Pr\left(|cos\theta_{x_i,x_j}| > t\right) < \frac{Var[x_i \cdot x_j]\frac{1}{n}}{t^2} \approx \frac{1}{nt^2} \tag{S.10}$$

. Note this holds as the datasets are independent with no shared information. However, if the datasets are not independent due to underlying common features, this bound no longer holds and we expect randomly drawn vectors to be non-orthogonal. Note similarly to Lemma 2, this orthogonality holds as $n \to \infty$. In particular, it must scale as a function of dataset size.

### S.3.2  NEURAL NETWORK LEARNING

### S.3.2.1  PRELIMINARIES

1. The activation function of a neural network can be denoted as $\phi(w \cdot x_i)$ where $\phi$ is the activation function, $w$ is the weight and $x_i$ is the feature vector.

2. In supervised learning, the loss function is typically a function of $\phi(w \cdot x_i)$ and the target $y_i$, $L(\phi(w \cdot x_i), y)$.

3. During training, the weights are updated through gradient descent, which adjusts the weights in the direction of the negative gradient of the loss function. This process is given by the equation $g = -\eta\nabla_w L(w_t, x_i, y_i)$, where $L(.,.,.)$ is the loss function, $\nabla_w$ denotes the gradients with respect to the weights, and $\eta$ is the learning rate. The gradient, $g$, is a measure of how the loss changes with to the weights. Alignment between the $g$ and $w$ is denoted by $g \cdot w$ and indicates the direction of change to minimize the loss.

4. Each input data point $x$ can be broken into two components, one that is parallel to the weight vector $w$, denoted $x_\parallel$, and one that is perpendicular, denoted $x_\perp$. Formally, we express this as $x = x_\parallel + x_\perp$, where $x_\parallel = \frac{(x \cdot w)w}{||w||^2} = (x \cdot \tilde{w})\tilde{w}$, and $x_\perp = x - x_\parallel$

5. The trigonometric identity for $cos(x) + cos(y)$ is $2cos(\frac{x+y}{2})cos(\frac{x-y}{2})$

### S.3.3  PROOF 1: GRADIENT IS PROPORTIONAL TO THE PARALLEL COMPONENT OF THE INPUT VECTOR

We can express $g$ in terms of the input vector, $g = \phi'(w \cdot x)x$ where $\phi'$ is the derivative of the activation function. Decomposing $x$ leads to $g = \phi'(w \cdot (x_\parallel + x_\perp))(x_\parallel + x_\perp) = \phi'(w \cdot x_\parallel)x_\parallel + \phi'(w \cdot x_\perp)x_\perp$ However,

as $w \cdot x_{\perp} = 0$, $g = \phi'(w \cdot x_{\parallel})x_{\parallel}$. Note in a multi-layered network we replace $g = \phi'(w \cdot x_{\parallel})x_{\parallel}$ with $g \propto \phi'(w \cdot x_{\parallel})x_{\parallel}$ as the feature vectors goes through additional layers. This leaves us with:

$$\Delta w \propto g \propto x_{\parallel} \tag{S.11}$$

*Remarks*:

- In deeper networks, this proportionality may only hold in the early layers. However, its been shown that early layers function to extract fundamental features from the dataset that are used by deeper layer Zeiler & Fergus (2014); Yosinski et al. (2015). Thus, the relationship will still have implications for a deeper network. Further, assuming neural networks acts as a mapping function from input space $X$ to output space $Y$ and that this function is Lipschitz continuous, we can invoke the smoothness assumption. Formally, $\exists K : \forall x_1, x_2 \in X \Rightarrow d_Y(f(x_1), f(x_2)) \leq K d_X(x_1, x_2)$. This essentially means that data points which are proximate in the feature space will continue to remain close in the output space, even when subjected to the non-linear transformations of the model

- Early during training most networks are initialized such that $w \cdot x$ is in a the linear region of the activation function to allow training He et al. (2015); Heaton (2018). While this may breakdown over training, the direction of early training has consequences for the final model.

### S.3.4    PROOF 2: THE TOTAL CHANGE IN $w$ IS PROPORTIONAL TO THE SUMMED PARALLEL COMPONENT OF ALL VECTORS

Without loss of generality, let's consider from our dataset $\mathcal{D}$ two vectors, $x_i$ and $x_j$ with losses $L_i$ and $L_j$, respectively. The total loss for these data points is $L = L_i + L_j$. Using the fact that the derivative of a sum is the sum of the derivatives, we get $\nabla_w L = \nabla_w L_i + \nabla_w L_j$. From our previous analysis we showed $\eta \nabla_w L_i \propto x_{i\parallel}$ such that:

$$\Delta w \propto x_{i\parallel} + x_{j\parallel} \tag{S.12}$$

This logic can be extended to $n$ number of data points. Note, we assume that the constant of proportionality is the same for both vectors. This constant is impacted by $\eta$ and $\phi'(.)$. For a given training round, $\eta$ remains the same for all vectors. Further, with random or He initialization of weights, the expected value of $\phi'(.)$ should be similar for both vectors.

### S.3.5    PROOF 3: THE TOTAL CHANGE IN $w$ IS PROPORTIONAL TO THE ANGLE BETWEEN FEATURE VECTORS

We use the previous result $\Delta w \propto x_{i\parallel} + x_{j\parallel}$ but replace $x_{i\parallel}$ with $cos(\theta_{x_i w})$ where $\theta_{x_i w}$ is the angle between the weight and feature vector. This is possible because the projection of $x_i$ onto $w$ (which is $x_{i\parallel}$) is equal to $||x_i||cos(\theta_{x_i w})$, where $||x_i||$ is the magnitude of $x_i$. Since we consider the proportionality, we can ignore the magnitude and focus on the cosine term. Then, $\Delta w \propto cos(\theta_{x_i w}) + cos(\theta_{x_j w}) = 2cos(\frac{\theta_{x_i w} + \theta_{x_j w}}{2})cos(\frac{\theta_{x_i w} - \theta_{x_j w}}{2})$. The first term is the cosine of the average angle between $w$ with $x_i$ and $x_j$ and represents alignment of each vector with $w$. The second term represents the cosine of half the difference in angles between $w$ with $x_i$ and $x_j$ and represents the alignment of the vectors to each other. This second term is maximized when the difference in angles is small. Under random initialization, the alignment of each vector with $w$ (*i.e.,* the cosine of the angle between each vector and $w$) is expected to be similar due to the symmetry of the initialization distribution. This means that the first term will be nearly constant and the second term, which represents the alignment of the vectors to each other, will dominate. This leaves us with:

$$\Delta w \propto cos(\theta_{x_i x_j}) \tag{S.13}$$

If the cosine angle is close to 1, then the change in $w$ will be greatest, leading to the largest improvement in the loss. This suggests that the alignment of feature vectors to each other has a role in neural network learning.

#### S.3.5.1    GRADIENT DIVERSITY

The equation for gradient diversity is:

$$\Delta_S(w) := \frac{\Sigma_{i=1}^n ||\nabla_i(w)||_2^2}{||\Sigma_{i=1}^n \nabla_i(w)||_2^2} \tag{S.14}$$

where $\nabla_i(w)$ is model gradients. Notably, gradient diversity tends to be higher when the inner products between the gradients are smaller.

## S.4 EXPERIMENTS

### S.4.1 SETUP: DATASETS, TRAINING, AND MODELS

#### S.4.1.1 SETUP: DATASETS

Table S.1 describes how we partitioned datasets to produce datasets with specific dataset dissimilarity scores. We employ a mixture of feature and label skew Hsieh et al. (2020); Zhao et al. (2018). Specifically:

- **Synthetic:** We draw samples from 2 distinct distributions, $\mathcal{D}_a$ and $\mathcal{D}_b$. For partition 1 we always draw samples from $\mathcal{D}_a$ and for partition 2 we vary the proportion of samples drawn from $\mathcal{D}_a$ and $\mathcal{D}_b$. The fewer samples in partition 2 drawn from $\mathcal{D}_a$, the greater the dissimilarity between the 2 partitions. We also include two additional illustrative examples to help with intuition of the score: identical datasets and when one dataset is a 50% subset of the other.

- **Credit:** We introduce increasing amounts of label skew and feature noise to the datasets. Label skew is achieved by introducing label distribution bias. For example, a 10% bias means 10% more positive labels and negative labels in parition 1 and 2, respectively. Feature noise is achieved by adding Gaussian noise.

- **Weather:** We partition datasets based on climate as described by Malinin et al. (2021). This introduces real-world feature and label skew. Note, the task is to predict the actual temperature, *i.e.,* a regression task.

- **EMNIST:** We partition datasets based on label overlap. We determine overlap using the *'class'* and *'merge'* splits. The 'class' split treats every lower and upper case letter as a distinct label. The 'merge' split combines labels for letters where the upper and lower case are visually similar *e.g.,* "i" and "I". At low dissimilarity, datasets share the same 'class' labels To increase dissimilarity, we reduce 'class' label overlap and increase 'merge' label overlap. For higher dissimilarity, we make labels discordant. Note that for training and prediction, the assigned label is always from the 'class' split.

- **CIFAR:** We partition the dataset based on labels, varying the number of overlapping labels in the two partitions. At low dataset dissimilarity scores, the datasets share many fine labels, meaning that they contain many images of the exact same type. As we increase dataset dissimilarity, we reduce the overlap of fine labels and increase the overlap of coarse labels, meaning that they contain more images of similar types *e.g.,* horse and zebra. To obtain the highest cost, we partition datasets giving them discordant labels.

- **IXITiny:** We partition based on where the images were collected. This is a natural partition. Sites Guys and Hammersmith use a Philips system with similar settings. Site IOP uses a GE system with unknown settings. See Brain-development.org; Terrail et al. (2022) for more details.

- **ISIC2019:** We partition based on where the images were collected. This is a natural partition. Note we split site ViDIR Group, Vienna into 3 groups based on the imaging machine used to collect the data as in Terrail et al. (2022). We aim to use ViDIR Group, Vienna as a fixed site and compare performance when it is combined with another site. We do this as ViDIR Group, Vienna allows us to explore the impact of combining data from different geographical sites and/or different imaging machines.

Table S.1: Creation of partitioned datasets

| Dataset | OT score | Partition 1 | Partition 2 |
|---|---|---|---|
| Synthetic | 0.00* | 100% $\mathcal{D}_a$ | Identical to Partition 1 |
| | 0.01* | 100% $\mathcal{D}_a$ | 50% subset of Partition 1 |
| | 0.03 | 100% $\mathcal{D}_a$ | 100% $\mathcal{D}_a$ |
| | 0.1 | 100% $\mathcal{D}_a$ | 85% $\mathcal{D}_a$, 15% $\mathcal{D}_b$ |
| | 0.2 | 100% $\mathcal{D}_a$ | 65% $\mathcal{D}_a$, 35% $\mathcal{D}_b$ |
| | 0.3 | 100% $\mathcal{D}_a$ | 43% $\mathcal{D}_a$, 57% $\mathcal{D}_b$ |
| | 0.4 | 100% $\mathcal{D}_a$ | 25% $\mathcal{D}_a$, 75% $\mathcal{D}_b$ |
| | 0.5 | 100% $\mathcal{D}_a$ | 11% $\mathcal{D}_a$, 89% $\mathcal{D}_b$ |
| Credit | 0.12 | No feature noise or label bias | |
| | 0.23 | No feature noise and 30% label bias | |
| | 0.3 | feature noise $\sim N(0.5, 0.5)$ and 45% label bias | |
| | 0.4 | feature noise $\sim N(1, 0.5)$ and 45% label bias | |
| Weather | 0.11 | Tropical, Mild temperate | |
| | 0.19 | Tropical, Mild temperate | Dry, Mild temperate |
| | 0.30 | Tropical, Mild temperate | Dry |
| | 0.40 | Tropical, Mild temperate | Dry, Snow |
| | 0.48 | Tropical, Mild temperate | Snow |
| EMNIST | 0.11 | 1-10 | |
| | 0.19 | 1-10, 12,18,24,28 | 1-10, 38,44,50,54 |
| | 0.25 | 1-10, 11,12,13,14,16,18,24,28 | 1-10, 37,38,39,40,42,44,50,54 |
| | 0.34 | 1-10, 11-25 | 1-10, 36-41 |
| | 0.39 | 1-10, 11-35 | 1-10, 36-61 |
| CIFAR | 0.08 | 1-10 | |
| | 0.21 | 11,98,29,73, 78, 49, 97, 51, 55, 92 | 11,98,29,73, 78, 49, 42, 83, 72, 82 |
| | 0.30 | 11,50,78,1,92, 78, 49, 97, 55, 16, 14 | 11, 36, 29, 73, 82, 78, 49, 42, 12, 23, 51 |
| | 0.38 | 11,50,78,8,92,2,49,98,89,3 | 17, 36, 30, 73, 83,28, 34, 42, 10, 20 |
| IXITiny | 0.08 | Guys | Hammersmith |
| | 0.28 | Guys | IOP |
| | 0.30 | Hammersmith | IOP |
| ISIC2019 | 0.06 | ViDIR Group, Vienna (FOTO) | ViDIR Group, Vienna (FOTO) |
| | 0.15 | ViDIR Group, Vienna (FOTO) | Hospital Clínic de Barcelona |
| | 0.19 | ViDIR Group, Vienna (FOTO) | ViDIR Group, Vienna (Dermaphot) |
| | 0.25 | ViDIR Group, Vienna (FOTO) | ViDIR Group, Vienna (MoleMax) |
| | 0.3 | ViDIR Group, Vienna (MoleMax) | Cliff Rosendahl, Australia |

* Note these examples are not assessed during model training and are for illustrative purposes only.

### S.4.1.2  SETUP: TRAINING AND MODELS

All models and hyperparameters are made available at `github.com/iclr-otcost-submission/submit`. For each combination of dataset, cost, and training, we performed learning rate and optimizer tuning via grid search. This encompasses five learning rates ($5e^{-1}, 1e^{-1}, 5e^{-2}, 1e^{-2}, 5e^{-3}, 5e^{-4}$) and two optimizers: ADAM and SGD. Note, the SGD optimizer is exclusively used for federated algorithms as it can improve convergence in some cases. For CIFAR, IXITiny and ISIC2019 we utilize pre-trained models that we fine-tune. The models selected are the best performing models that are publicly available.

**Synthetic**    We use a MLP with 2 hidden layers of size 18 and 6. We use BCE loss, ReLU activation, and dropout of 0.3. We use the following learning rates, optimizers, and regularization parameters (for pfedme and Ditto only):

Table S.2

| OT score | Single | Joint | FedAvg | pfedme | Ditto |
|---|---|---|---|---|---|
| 0.03 | $5e^{-3}$, ADM | $5e^{-2}$, ADM | $5e^{-2}$, ADM | $5e^{-2}$, ADM, $5e^{-1}$ | $1e^{-1}$, ADM, $5e^{-1}$ |
| 0.10 | $5e^{-3}$, ADM | $5e^{-2}$, ADM | $1e^{-1}$, ADM | $1e^{-1}$, ADM, $5e^{-1}$ | $1e^{-1}$, ADM, $1e^{0}$ |
| 0.20 | $1e^{-2}$, ADM | $1e^{-1}$, ADM | $1e^{-1}$, ADM | $1e^{-1}$, ADM, $1e^{-1}$ | $1e^{-1}$, ADM, $1e^{-1}$ |
| 0.30 | $5e^{-2}$, ADM | $5e^{-3}$, ADM | $5e^{-2}$, SGD | $5e^{-2}$, SGD, $1e^{-2}$ | $1e^{-1}$, SGD, $1e^{-2}$ |
| 0.40 | $5e^{-2}$, ADM | $1e^{-2}$, ADM | $5e^{-2}$, SGD | $5e^{-2}$,SGD, $1e^{-2}$ | $1e^{-1}$, SGD, $1e^{-2}$ |
| 0.50 | $5e^{-2}$, ADM | $3e^{-2}$, ADM | $5e^{-2}$, SGD | $1e^{-1}$, SGD, $1e^{-3}$ | $1e^{-1}$, SGD, $1e^{-3}$ |

**Credit**    We use a MLP with 3 hidden layers of size 56, 56 and 28. We use BCE loss, ReLU activation, batch normalization and dropout of 0.5. We use the following learning rates, optimizers, and regularization parameters (for pfedme and Ditto only):

Table S.3

| OT score | Single | Joint | FedAvg | pfedme | Ditto |
|---|---|---|---|---|---|
| 0.12 | $5e^{-3}$, ADM | $5e^{-2}$, ADM | $5e^{-2}$, ADM | $1e^{-1}$, ADM, $5e^{-1}$ | $1e^{-1}$, ADM, $1e^{0}$ |
| 0.23 | $5e^{-3}$, ADM | $1e^{-2}$, ADM | $1e^{-2}$, ADM | $1e^{-1}$, ADM, $1e^{-1}$ | $1e^{-1}$, ADM, $1e^{-1}$ |
| 0.30 | $1e^{-2}$, ADM | $1e^{-2}$, ADM | $1e^{-2}$, ADM | $1e^{-1}$, ADM, $1e^{-2}$ | $1e^{-1}$, ADM, $1e^{-3}$ |
| 0.40 | $5e^{-2}$, ADM | $1e^{-3}$, ADM | $5e^{-3}$, ADM | $1e^{-1}$, ADM, $1e^{-2}$ | $1e^{-1}$, ADM, $1e^{-3}$ |

**Weather**    We use a MLP with 3 hidden layers of size 123, 123 and 50. We use MSE loss, ReLU activation, batch normalization and dropout of 0.5. We use the following learning rates, optimizers, and regularization parameters (for pfedme and Ditto only):

Table S.4

| OT score | Single | Joint | FedAvg | pfedme | Ditto |
|---|---|---|---|---|---|
| 0.11 | $3e^{-3}$, ADM | $1e^{-2}$, ADM | $5e^{-2}$, SGD | $1e^{-1}$, ADM, $5e^{-1}$ | $1e^{-1}$, ADM, $1e^{0}$ |
| 0.19 | $5e^{-3}$, ADM | $5e^{-2}$, ADM | $1e^{-1}$, SGD | $1e^{-1}$, ADM, $5e^{-1}$ | $1e^{-1}$, ADM, $1e^{0}$ |
| 0.30 | $1e^{-2}$, ADM | $1e^{-2}$, ADM | $5e^{-2}$, SGD | $1e^{-1}$, ADM, $1e^{-1}$ | $1e^{-1}$, ADM, $5e^{-1}$ |
| 0.40 | $1e^{-2}$, ADM | $1e^{-3}$, ADM | $2e^{-2}$, SGD | $1e^{-1}$, ADM, $1e^{-2}$ | $1e^{-1}$, ADM, $1e^{-2}$ |
| 0.48 | $1e^{-2}$, ADM | $1e^{-3}$, ADM | $5e^{-2}$, SGD | $1e^{-1}$, ADM, $1e^{-3}$ | $1e^{-1}$, ADM, $1e^{-2}$ |

**EMNIST**    We use a LeNet-5 CNN. We use CrossEntropy loss, ReLU activation, and batch normalization. We use the following learning rates, optimizers, and regularization parameters (for pfedme and Ditto only):

Table S.5

| OT score | Single | Joint | FedAvg | pfedme | Ditto |
|---|---|---|---|---|---|
| 0.11 | $5e^{-2}$, ADM | $1e^{-2}$, ADM | $5e^{-2}$, ADM | $5e^{-2}$, ADM, $1e^{-1}$ | $5e^{-2}$, ADM, $5e^{-1}$ |
| 0.19 | $5e^{-3}$, ADM | $5e^{-3}$, ADM | $5e^{-2}$, ADM | $5e^{-2}$, ADM, $1e^{-1}$ | $5e^{-2}$, ADM, $1e^{-1}$ |
| 0.25 | $1e^{-2}$, ADM | $5e^{-3}$, ADM | $5e^{-2}$, ADM | $5e^{-2}$, ADM, $1e^{-2}$ | $1e^{-1}$, ADM, $1e^{-2}$ |
| 0.34 | $1e^{-2}$, ADM | $1e^{-2}$, ADM | $1e^{-2}$, ADM | $5e^{-2}$, ADM, $1e^{-2}$ | $1e^{-1}$, ADM, $1e^{-3}$ |
| 0.39 | $1e^{-2}$, ADM | $5e^{-3}$, ADM | $3e^{-2}$, ADM | $1e^{-1}$, ADM, $1e^{-3}$ | $5e^{-2}$, ADM, $1e^{-3}$ |

**CIFAR**    We use a pre-trained ResNet-18. We freeze layers 1-3 and allow training of layer 4 and the fully connected layer. We use CrossEntropy loss. We use the following learning rates, optimizers, and regularization parameters (for pfedme and Ditto only):

Table S.6

| OT score | Single | Joint | FedAvg | pfedme | Ditto |
|---|---|---|---|---|---|
| 0.08 | $1e^{-3}$, ADM | $5e^{-4}$, ADM | $1e^{-2}$, ADM | $1e^{-2}$, ADM, $5e^{-1}$ | $5e^{-3}$, ADM, $1e^{0}$ |
| 0.21 | $1e^{-3}$, ADM | $5e^{-3}$, ADM | $1e^{-2}$, SGD | $5e^{-2}$, SGD, $1e^{-1}$ | $1e^{-2}$, ADM, $5e^{-1}$ |
| 0.30 | $5e^{-4}$, ADM | $5e^{-2}$, ADM | $1e^{-2}$, SGD | $5e^{-2}$, SGD, $1e^{-2}$ | $1e^{-2}$, ADM, $1e^{-2}$ |
| 0.38 | $5e^{-4}$, ADM | $5e^{-3}$, ADM | $5e^{-2}$, SGD | $5e^{-2}$, SGD, $1e^{-3}$ | $5e^{-3}$, ADM, $1e^{-2}$ |

**IXITiny**  We use a pre-trained 3D Unet and allow training of all layers. We use DICE loss. We use the following learning rates, optimizers, and regularization parameters (for pfedme and Ditto only):

Table S.7

| OT score | Single | Joint | FedAvg | pfedme | Ditto |
|---|---|---|---|---|---|
| 0.08 | $1e^{-1}$, ADM | $1e^{-1}$, ADM | $5e^{-2}$, ADM | $1e^{-1}$, ADM, $5e^{-1}$ | $1e^{-1}$, ADM, $1e^{-1}$ |
| 0.28 | $1e^{-1}$, ADM | $1e^{-1}$, ADM | $1e^{-1}$, SGD | $5e^{-2}$, SGD, $1e^{-2}$ | $1e^{-1}$, SGD, $1e^{-2}$ |
| 0.30 | $1e^{-1}$, ADM | $1e^{-2}$, ADM | $1e^{-1}$, SGD | $5e^{-2}$, SGD, $1e^{-3}$ | $1e^{-3}$, SGD, $1e^{-2}$ |

**ISIC2019**  We use a pre-trained efficientNet and allow training of all layers. We use CrossEntropy loss. We use the following learning rates, optimizers, and regularization parameters (for pfedme and Ditto only):

Table S.8

| OT score | Single | Joint | FedAvg | pfedme | Ditto |
|---|---|---|---|---|---|
| 0.06 | $5e^{-4}$, ADM | $5e^{-3}$, ADM | $1e^{-1}$, ADM | $5e^{-3}$, ADM, $5e^{-1}$ | $1e^{-2}$, ADM, $5e^{-1}$ |
| 0.15 | $5e^{-3}$, ADM | $5e^{-3}$, ADM | $1e^{-2}$, ADM | $5e^{-3}$, ADM, $1e^{-1}$ | $1e^{-1}$, ADM, $1e^{-1}$ |
| 0.19 | $5e^{-3}$, ADM | $5e^{-3}$, ADM | $1e^{-2}$, ADM | $5e^{-3}$, ADM, $1e^{-1}$ | $1e^{-1}$, ADM, $1e^{-1}$ |
| 0.25 | $5e^{-3}$, ADM | $1e^{-2}$, ADM | $1e^{-2}$, ADM | $1e^{-2}$, ADM, $1e^{-2}$ | $1e^{-2}$, ADM, $1e^{-2}$ |
| 0.3 | $5e^{-3}$, ADM | $5e^{-3}$, ADM | $5e^{-2}$, ADM | $1e^{-2}$, ADM, $1e^{-3}$ | $1e^{-2}$, ADM, $1e^{-2}$ |

*ADM = Adam optimizer, SGD = SGD optimizer

## S.5 RESULTS

### S.5.1 RESULTS: ESTIMATES USING PRIVACY-PRESERVING SMPC CALCULATION

Figure S.1 compares the calculated score from the plain-text algorithm and the algorithm that leverages SMPC to calculate scores in a privacy-preserving way. We show that the SMPC protocol introduces no error ($\leq 1e^{-6}$) in calculating our metric.

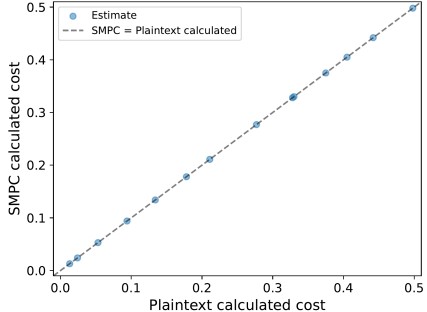

Figure S.1: **Scores calculated in plaintext vs. calculated using SMPC**. Dashed Y=X represents perfect agreement.

### S.5.2 RESULTS: ESTIMATES USING PRIVACY-PRESERVING DP CALCULATION

Figure S.2 compares the calculated label costs without and with zCDP noise added using COINPRESS Biswas et al. (2020). We show that the DP protocol introduces minimal error across most label comparisons.

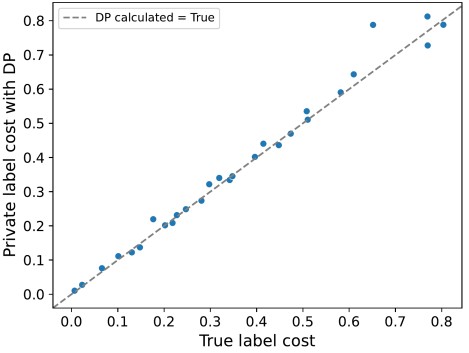

Figure S.2: **Scores calculated without vs. with zCDP noise**. Dashed Y=X represents perfect agreement.

### S.5.3 RESULTS: COMPUTATIONAL AND SAMPLE SIZE COMPLEXITY

We provide a comparison between the dataset dissimilarity scores derived from the complete datasets and those obtained from a randomly subsampled subset (Fig S.3). Notably, even though sample complexity grows with dataset dimensionality, our observations suggest that roughly 1000 samples suffice to yield precise estimates for datasets of the highest dimensions. The theoretical bounds OT sample complexity using Wasserstein distance has been explored in Mena & Niles-Weed (2019); Genevay et al. (2019). As sample complexity is generally tied to the learning algorithm rather than the metric itself, we believe similar bounds hold for our formulation too.

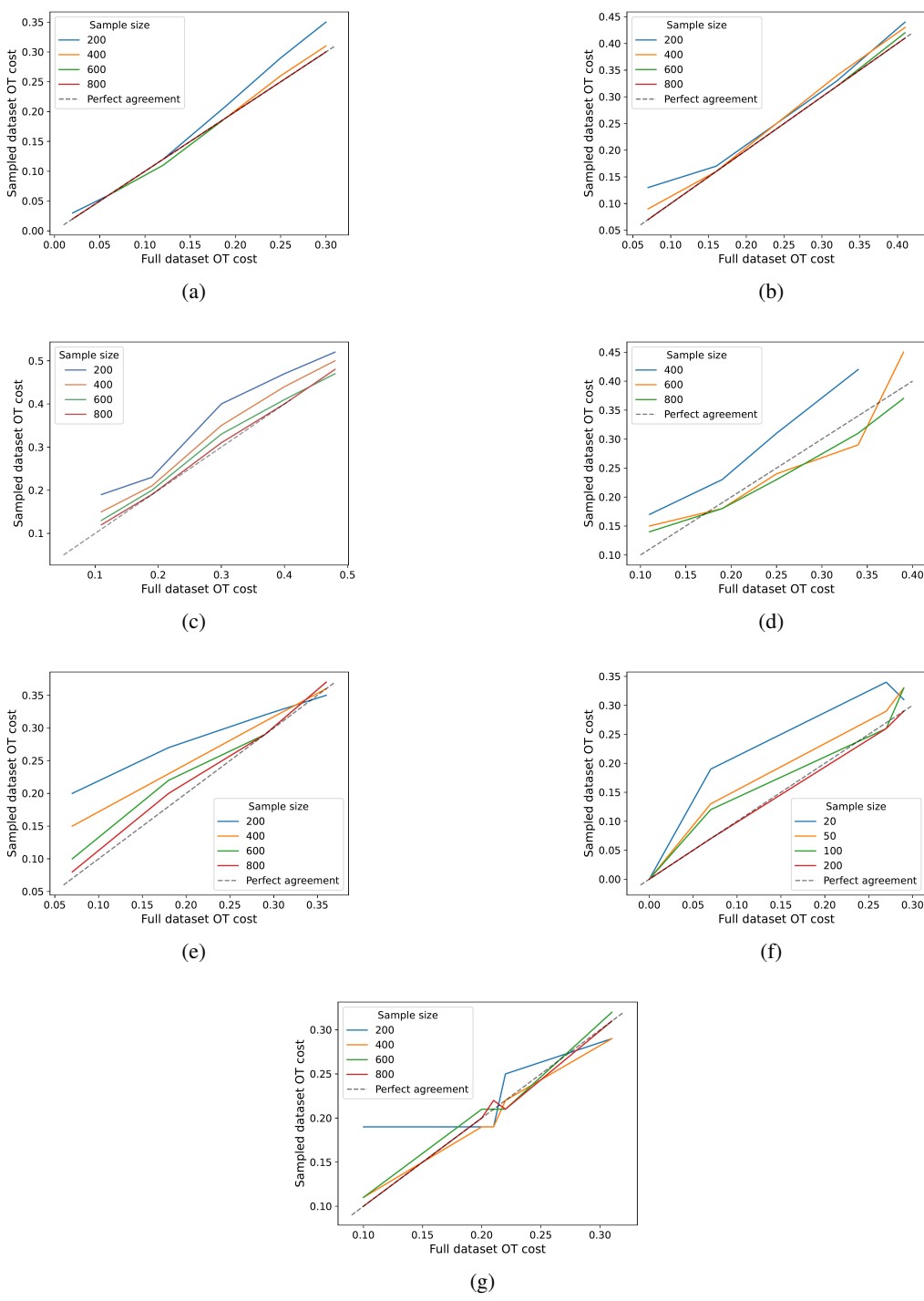

Figure S.3: **Dataset dissimilarity scores calculated from full dataset and subsampled dataset**. Dashed Y=X represents perfect agreement (a) synthetic, (b) credit (b), weather (c), EMNIST (d), CIFAR (e), IXITiny (f), and ISIC2019 (g).

### S.5.4   RESULTS: ADVANTAGES OF OUR METRIC OVER TRADITIONAL MEASURES

#### S.5.4.1   WASSERSTEIN DISTANCE

We evaluate our metric against the conventional Wasserstein distance using the method introduced by Alvarez-Melis & Fusi (2020) (Fig.S.4). Importantly, the score leveraging Wasserstein distance is dataset scale and

dimension dependent making consistent interpretation of the score across different datasets challenging *e.g.,* the same score leads to improved learning in IXITiny but not in ISIC2019.

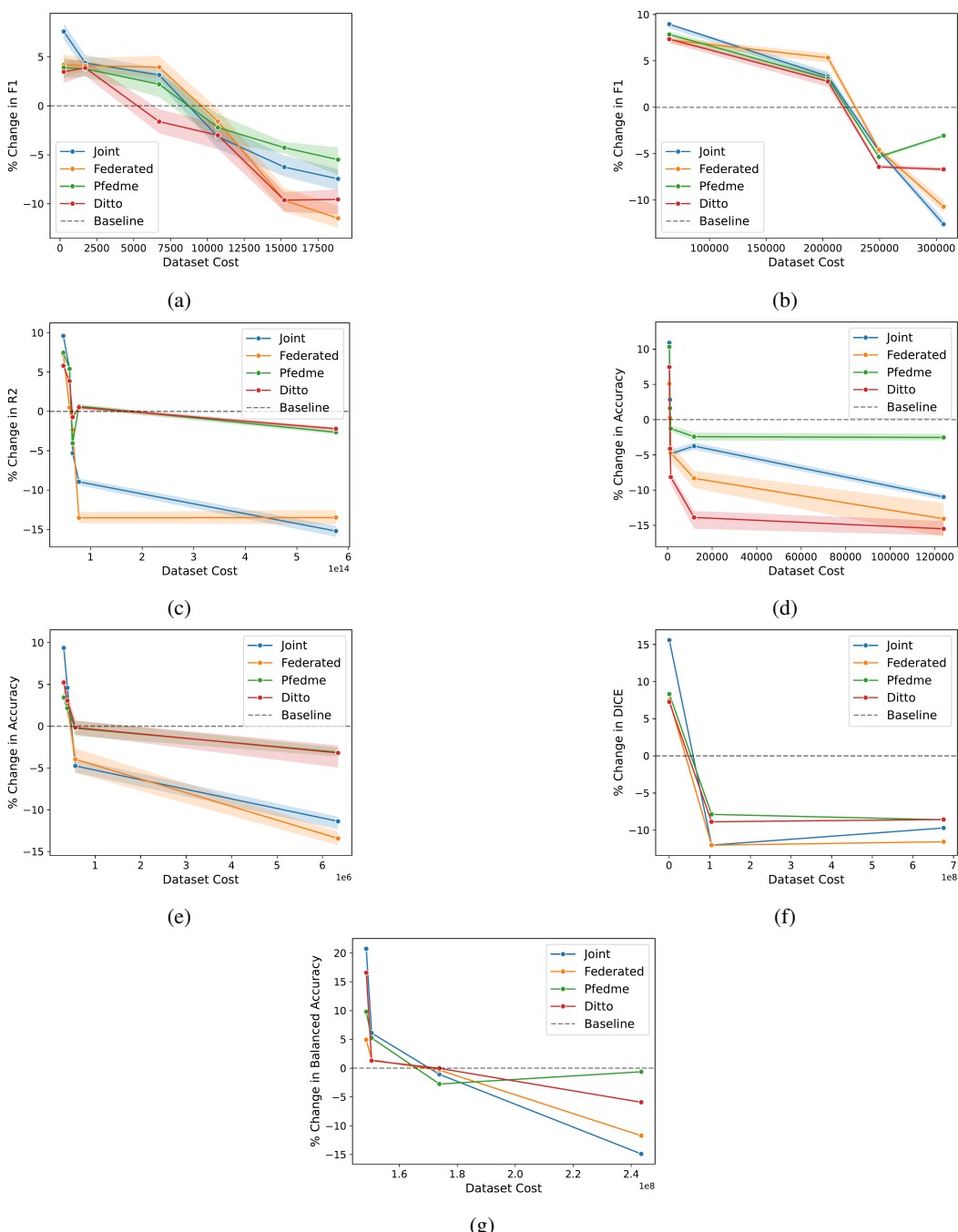

Figure S.4: **Performance of models trained on datasets with different wasserstein dataset dissimilarity scores**. Results expressed as % change from single model baseline with Joint (blue), Federated (orange), pfedme (green) and Ditto (Red). Results shown for synthetic dataset (a) synthetic, (b) credit (b), weather (c), EMNIST (d), CIFAR (e), IXITiny (f), and ISIC2019 (g).

.

### S.5.4.2 RESULTS: SYNTHETIC RESULTS FOR COSTS > 0.5

Figure S.5 presents the results obtained from synthetic datasets with scores of 0.6, 0.8, and 1.0. To generate these datasets, we employed a combination of label switching and feature distributions that are deliberately contrasting. While we acknowledge that such a scenario may not directly mirror real-world conditions, we believe it helps with intuition of our score.

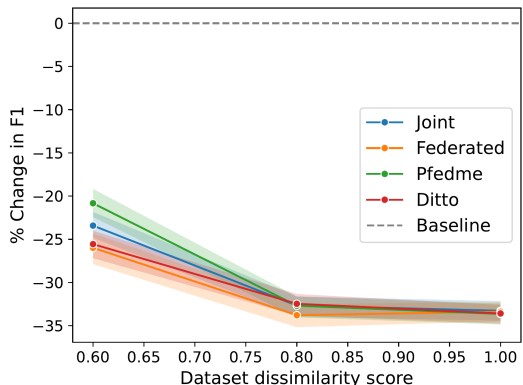

Figure S.5: **Performance of models trained on Synthetic dataset with scores 0.6, 0.8, 1.0**. Results expressed as percentage changes from single baseline with Joint (blue), Federated (orange), pfedme (green) and Ditto (Red)

### S.5.4.3 GRADIENT DIVERSITY CORRELATION

We assess the relationship between gradient diversity and two metrics: our proposed metric and the standard Wasserstein distance. This assessment is conducted using Pearson correlation tests, with the results detailed in Table S.9. Our findings indicate that our metric consistently exhibits a higher correlation with gradient diversity across all datasets when compared to the Wasserstein distance. To establish the statistical significance of this observation, we employ a paired Student's t-test, yielding a t-statistic of 2.87 and a p-value of 0.028.

Table S.9: Comparison of Pearson Correlation Coefficients: Gradient Diversity vs. Our Metric and Wasserstein Distance Across All Datasets

| Dataset | Our metric | Wasserstein |
|---------|------------|-------------|
| Synthetic | 0.83 | 0.80 |
| Credit | 0.92 | 0.84 |
| Weather | 0.76 | 0.11 |
| EMNIST | 0.78 | 0.30 |
| CIFAR | 0.95 | 0.57 |
| ISIC | 0.99 | 0.97 |
| IXITiny | 0.99 | 0.77 |

### S.5.4.4 KL DIVERGENCE

We also compare our metric to KL-divergence (Fig.S.6). To do this we simulate a small synthetic dataset with 2 dimensions. The reason is that at high dimension the sample complexity of KL-divergence is too large and accurate estimates could not be obtained. In two dimensions we find that our metric provides a more consistent and linear relationship with model performance. Conversely, KL divergence relationship to model performance appears complex and hard to interpret.

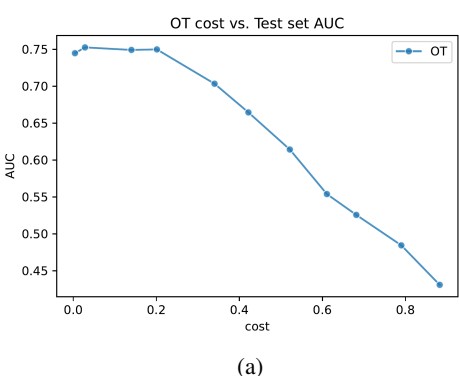 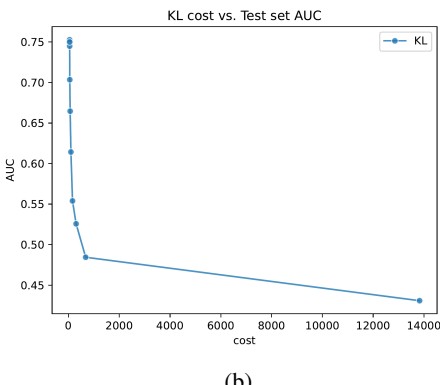

(a) (b)

Figure S.6: **Metric scores for synthetic 2D dataset** (a), OT metric (ours) (b), KL-divergence

### S.5.4.5 SMPC RUNTIME COMPLEXITY

**Dot product** Using the protocol outline by Du et al. (2004), the runtime complexity of the dot product protocol is $O(n^2 d + d^2)$. Here, $n$ is the number of samples and $d$ is the vector dimension. The $O(d^2)$ term arises from inversion of an MDS code matrix Jog (2004). The dominating term in the complexity depends on the relative size of sample count, $n$, to the dataset dimension, $d$.

**Euclidean distance** We also analyzed a protocol by Ravikumar et al. (2004) to calculate euclidean distance in SMPC. We chose this protocol as it also does not require encryption making it computationally efficient. The protocol provides a probabilistic estimate of the euclidean distance based on the number of indices from each vector that is used in the secure intersection step of the protocol. If one desires an exact euclidean distance estimate, then all indices must be sampled and the complexity is $O(n^2(d + dlog(d))$ where $n$ is the number of samples and $d$ is the vector dimension. Note if a non-exact estimate is sufficient, $dlog(d)$ can be replaced with $slog(s)$ where $s < d$.

### S.5.5 RESULTS: HYPERPARAMETERS

### S.5.5.1 REGULARIZATION PARAMETER $\epsilon$

In the Sinkhorn algorithm, the term $\epsilon$ is a weight for the entropic regularization term $H(\pi|X \otimes Y)$. Incorporating this regularization makes the optimization problem convex, making it solvable and reducing computational complexity from cubic to quadratic. However, there's a trade-off: as $\epsilon$ increases, the solution becomes closer to a uniform, and suboptimal, coupling. We conducted experiments across a range of values, $\epsilon = 5e^{-3}, 1e^{-2}, 5e^{-2}, 1e^{-1}, 5e^{-1}, 1, 10, 100, 1000, 10000$, and compared the scores obtained to the baseline $\epsilon = 1e^{-3}$. Our findings indicate that at values below $5^e-2$ the results are largely consistent (Fig S.7). However, $\epsilon$ becomes larger than this, the score tends to 0.5 suggesting uniform coupling. Based on our experiments, we recommend using values below $5^e-2$.

### S.5.5.2 FEATURE-TO-LABEL RATIO ($\lambda$)

We conduct a sensitivity analysis on the feature-to-cost label ratio. Our baseline, 2:1, arises from the fact that cosine similarity and Hellinger distance range from 0-2 and 0-1, respectively. We compare the following ratios: 1:1, 1:2, 1:3, 1:4, 1:5, 3:1, 4:1, 5:1. Our results show that the score is robust to a range of ratios within the 1:4-4:1 range suggesting there is general agreement between the label and feature costs (Fig S.8. Notably, we observed that increased label weights result in higher scores, while greater feature weights correspond to lower scores. This divergence becomes pronounced outside the 4:1 to 1:4 ratio range. While adjusting the ratio offers flexibility, we recommend maintaining it within the 4:1 to 1:4 range for optimal outcomes.

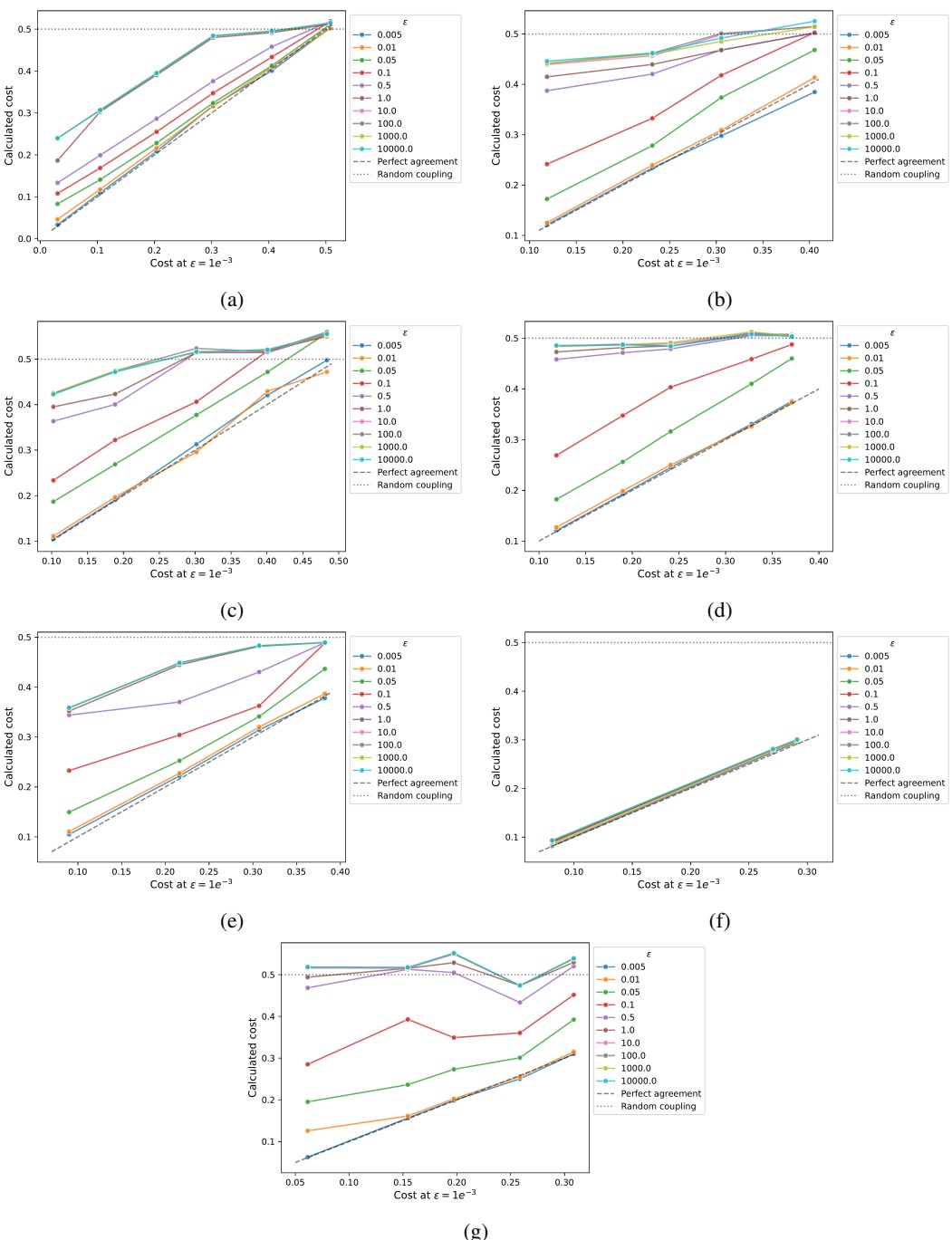

Figure S.7: **Scores calculated at varying $\epsilon$ values vs. scores calculated at** $\epsilon = 1e^{-3}$. Dashed Y=X represents perfect agreement with $\epsilon = 1e^{-3}$ (a) synthetic, (b) credit (b), weather (c), EMNIST (d), CIFAR (e), IXITiny (f), and ISIC2019 (g).

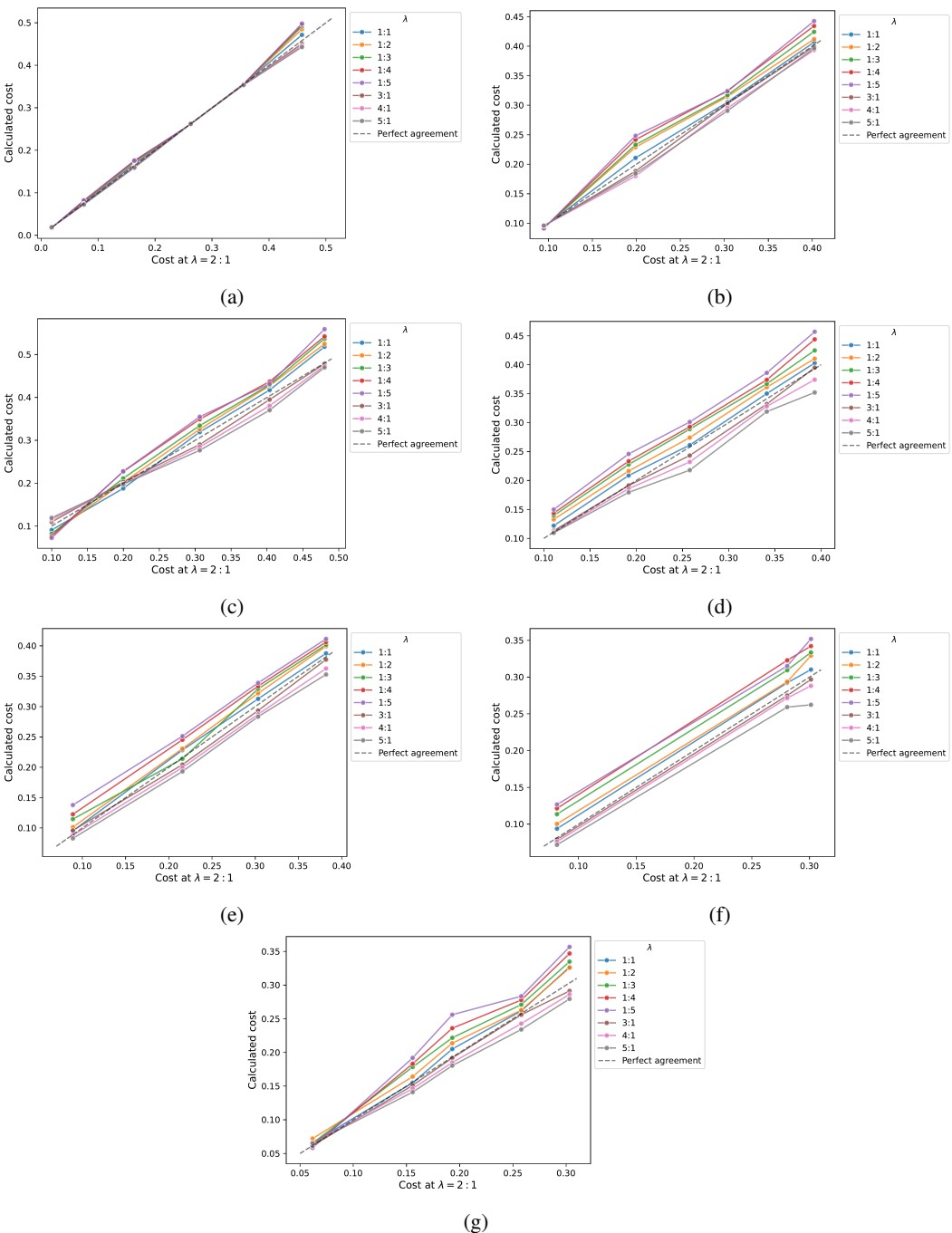

(a)

(b)

(c)

(d)

(e)

(f)

(g)

Figure S.8: **Scores calculated at varying feature-to-label cost ratios**. Dashed Y=X represents perfect agreement with 2:1 ratio (a) synthetic, (b) credit (b), weather (c), EMNIST (d), CIFAR (e), IXITiny (f), and ISIC2019 (g).