# OpenReview forum: "A universal metric of dataset similarity for multi-source learning"
_ICLR.cc/2024/Conference — Submitted to ICLR 2024_

### Official Review · Reviewer_U4Pq · 2023-10-31

**Soundness:** 2 fair
**Presentation:** 2 fair
**Contribution:** 2 fair
**Rating:** 5
**Confidence:** 3

**Summary:**

This paper developed a metric to quantify the similarity between two datasets. To obtain the metric, the first step is computing cosine similarity between pairs of data samples, added by Hellinger distance between the same pair of labels. Then, we obtain the distance between the two datasets by applying optimal transport over the cosine and Hellinger distances.

**Strengths:**

This paper studies a foundational problem in machine learning and may have a broad impact on many areas.

**Weaknesses:**

1. The paper first constructs a metric between a pair of data samples using cosine similarity and Hellinger distance. However, the Hellinger distance is not well motivated. What's the key advantage of the Hellinger distance over the Wasserstein distance?
2. The theoretical insights (Section 6) need improvements. The authors consider a scenario where "any pair of vectors drawn from two random and independent datasets". Such a scenario may not be representative enough because two different datasets may have overlaps. Also, the Hellinger distance is not discussed in the theoretical analysis.
3. I do not believe " estimating gradients similarity without model training" is a feature of the proposed approach. Deriving a bound of gradient similarity using data similarity is straightforward for a Lipschitz function. In this sense, any metric quantifying data similarity can estimate gradient similarity.

**Questions:**

N/A.

---

> ### Author Response · Authors · 2023-11-16
>
> ***the Hellinger distance is not well motivated. What's the key advantage of the Hellinger distance over the Wasserstein distance?***...***Also, the Hellinger distance is not discussed in the theoretical analysis***
>
> We thank the reviewer for this observation. We recognize the need for a clearer motivation behind our choice of the Hellinger distance. Our primary reason for using the Hellinger distance is to leverage a bounded cost function that is admissible with the optimal transport framework. Specifically, the Wasserstein distance is a special case of the (f, Γ)-divergence family where the Γ cost function is bounded 1-Lipschitz functions (e.g., L1 or L2 norm) (*Birrell et al., JMLR, 2022*). However, the (f, Γ)-divergence family framework inherently allows for flexibility in choosing alternate  Γ functions provided it possesses certain properties. Given we use the Hellinger Distance to generate the cost map, and the Hellinger Distance uses a bounded, continuous function, it can be substituted. The advantage of this modification is that it shifts interpretation from assessing distribution differences in Euclidean geometry to comparing probability densities within the manifold. This is because the Hellinger distance can be interpreted as representing the shortest geodesic distance in the statistical manifold. This approach provides a meaningful way to compare label distributions, preserving the optimal transport interpretation, while ensuring boundedness and relevance in probability space. We now include this more detailed rationale for Hellinger distance in *Section 6.2*.
>
> ***The authors consider a scenario where "any pair of vectors drawn from two random and independent datasets". Such a scenario may not be representative enough because two different datasets may have overlaps.***
>
> We thank the reviewer for this feedback. Our discussion focuses on scenarios involving pairs of vectors drawn from two random and independent datasets as this assumption allows us to produce our orthogonality bound. However, we state that this bound of orthogonality does not hold when datasets have overlap because we can no longer assume independence of the sample means and covariances (*Section S.3.1.4*). This bound and when it is violated motivates our primary reason for using the cosine similarity, i.e., that as datasets become increasingly less similar, more vectors are found to be orthogonal. Our empirical findings align with this argument. We have updated *Section 6.1* to make this assumption and when it is violated clearer.
>
> ***I do not believe " estimating gradients similarity without model training" is a feature of the proposed approach. Deriving a bound of gradient similarity using data similarity is straightforward for a Lipschitz function. In this sense, any metric quantifying data similarity can estimate gradient similarity.***
>
> We thank the reviewer for this comment. We would like to clarify that the assertion is not that our metric estimates gradient similarity without model training. Rather, our claim centers on the correlation between our metric and gradient similarity, and how this relationship can explain our metric's ability to estimate model performance without the need for explicit model training. While any Lipschitz continuous function can bound gradient similarity, the effectiveness of our metric lies in its stronger correlation. We attribute this to our specific choice of distances (cosine similarity + Hellinger distance) as we believe they are more closely aligned with the dynamics of neural network training. To test this empirically, we compared the correlation between gradient diversity and two metrics: our proposed metric and the standard Wasserstein distance. Our findings reveal that our metric consistently exhibits a higher correlation with gradient similarity compared to the Wasserstein distance across all datasets (with a statistically significant paired t-test, p=0.028, see *Table S.9*). This empirical evidence suggests that the modifications we have introduced to the conventional Wasserstein distance enhance its ability to estimate model performance. Note, in *Section S5.4.1* we already demonstrate how our metric aligns more closely with model performance compared to the Wasserstein distance.

---

> ### Comment · Reviewer_U4Pq · 2023-11-20
> **Reply to Authors**
>
> Thanks for replying. There are some interesting ideas in the comments, such as the specific choice of distances (cosine similarity + Hellinger distance) may be more closely aligned with the dynamics of neural network training. However, they are not very well justified in such a limited space.
>
> Therefore, I hope the authors could carefully revise this work and provide solid technical support to the claims.

---

> > ### Author Response · Authors · 2023-11-20
> >
> > We thank the reviewer for their response. We have revised our manuscript to provide more robust theoretical and empiricial support for our claims regarding the choice of distances. In particular we direct you to *"Section 6.1 UNDERSTANDING COSINE SIMILARITY AS AN APPROXIMATION FOR DATASET SIMILARITY IN DEEP LEARNING"* and *"Section 6.2 MOTIVATING THE USE OF HELLINGER DISTANCE"* for a theoretical justification for the use of the measures. Further, in *"Section S.5.4.3 GRADIENT DIVERSITY CORRELATION"* we empirically show that our metric has a stronger correlation with gradient diversity than the standard Wasserstein distance. We believe the theoretical justification and experimental validation provide a strong basis for our claims.

---

### Official Review · Reviewer_8TNG · 2023-11-06

**Soundness:** 3 good
**Presentation:** 3 good
**Contribution:** 2 fair
**Rating:** 5
**Confidence:** 3

**Summary:**

This paper studies the multi-source learning setting where different datasets may be non-identically-distributed. This paper improves on prior work by proposing a metric that satisfies many useful properties simultaneously: being bounded, being applicable to supervised learning (via accounting for the label distribution) and not requiring model training. The metric is based on optimal transport, cosine similarity, and hellinger distance. Experimental evaluation focuses on showing that this metric correlates with model performance as data is made more non-iid and with gradient diversity. This work also shows how to compute metrics inn SMPC setting.

**Strengths:**

This work has several strengths.

First, this metric is the first to achieve many desired quantities simultaneously. Though this work does so by using techniques already studied in machine learning (e.g., cosine distance, hellinger distance), it does combine these in a new way that leads to this benefit.

Second, this work is mostly clear and well written. For example, algorithm 1 clearly shows the cost metric and how it is computed in various settings. The notation is clear and easy to follow. There is also sufficient description of how to interpret the metric (e.g., how values of > 0.5 induce negative learning).

Third, there is sufficient related works and background. This makes it easy to understand the key contributions and placement in the literature, as well as interpret/understand the results.

Fourth, there are many experiments, including synthetic and real datasets covering many different cases (regression, multi-class image classification, etc.). The full details are also included enabling reproducibility. The results show a correlation between the metric and the desired quantity being measured: learning performance under varying degrees of non-iid datasets.

**Weaknesses:**

The first weakness is that this work claims privacy-preserving computation as a main contribution. However, this contribution is not clear, lacks any significant treatment of the techniques used in the main-text, and, is also missing important analysis. On clearness, this work claims to enable "privacy-preserving" computation many times early in the paper (abstract, fourth main contribution, to name two). This is vague. Does this mean DP, SMPC, or something else? This only becomes clear on the fourth page of the paper when the work first mentions SMPC. Importantly, though, this SMPC contribution is rather limited. On treatment and analysis, it does not include computation analysis, a security proof, and, appears to only use SMPC for the features (and not labels) as observed in Algorithm 1. This conflates the true security guarantees with what is provided as the work claims to provide a "privacy-preserving method for the metric". Further, this work also claims to provide a method that introduces no error, but the linked proof is actually a background (supplement S.1).

The second weaknesses is the empirical performance of the metric. Though there is certainly a correlation, this correlation appears to not be too strong in that it only well separates settings of IID ( where the metric is around 0.1 or lower) to those of heavily non-iid (where the metric is 0.3 or higher). In between this, there is large variation where the metric does not well correlate and has high variance in results across datasets. That being said, this result may be useful in itself, and so, this weakness is not major.

Third, there is lacking exploration with respect to non-iid learning approaches. The result that these approaches can impact utility on iid settings is interesting, however, this work also seems to show that these non-iid approaches often perform worse even in noniid settings. This is counter to their design and requires more exploration. Is there a reason that this is occurring?

NITs:

Supplemental broken citation at top of page 15.

**Questions:**

See Weaknesses.

---

> ### Author Response · Authors · 2023-11-16
>
> **Please note: A revised manuscript has been submitted which we reference in our response**
>
> ***claims privacy-preserving computation as a main contribution...***
>
> Thank you for your valuable feedback. We realize that our initial presentation of the privacy-preserving methods was not as clear as it could have been. Our primary contribution lies in developing a cost metric that can be calculated in a privacy-preserving way with little computational overhead (see *Section S.5.4.3*). This is desirable for many practitioners in distributed settings where privacy constraints limit multi-source learning. We feel this is an important contribution as, to our knowledge, other dataset similarity measures do not support privacy-preserving calculation. In particular, methods that use model training are not feasible using cryptographic techniques due to the performance overhead. In addition,  differential privacy can lead to considerable performance degradation when applied to complex models (*Bagdasaryan et al., NeurIPS, 2019*). In response to your feedback, we have revised the manuscript to more accurately reflect the scope of our contribution in this area. We also provide clearer guidance on where to find a comprehensive discussion of the methods.
>
> ***this SMPC contribution is rather limited.***
>
> We recognize that the treatment of SMPC in our manuscript does not include a security proof. Since our approach leverages existing methods, we omitted full handling of security guarantees as these have been shown elsewhere. We now make it clearer in the manuscript where such security guarantees can be found.
>
> Regarding computation analysis, we have provided this in *Section S.4.4.3* where we compare the computational cost for the dot product and comparison against the computational cost of Euclidean distance. This analysis demonstrates the feasibility and efficiency of our approach in practical scenarios. We have revised the manuscript accordingly.
>
> ***only use SMPC for the features (and not labels)***
>
> We thank the reviewer for this comment. We initially provided a method for the features as this required sharing of sample-level information whereas label costs are calculated on summary statistics. However, we acknowledge that in the low sample size setting, summary statistics can also leak information. In response to your feedback, we have revised our approach to ensure that both features and labels are treated with appropriate privacy-preserving methods. For labels, we use a zero-Concentrated Differential Privacy (ρ-zCDP, an intermediate method between pure and approximate DP) method that enables release of the summary statistics (*Biswas et al., 2020, NeurIPS*) . This method was chosen as it is computationally light and has sample complexity similar to the non-privacy-preserving methods. We show that under a strong privacy guarantee, ρ=0.1 or ε-δ= 0.99,0.01, (*Near and Abuah, Programming Differential Privacy, 2021*) we  retain high accuracy in the label cost calculation as we are able to provide the true sample measures a priori (see *Figure S.5.2*). We have amended *Algorithm 1* and added *Section S.2.3* describing how we apply this method in more detail.
>
> ***provide a method that introduces no error***
>
> We thank the reviewer for catching this typo. The manuscript should point to *Section S.2.2* for a theoretical proof and *S.5.1* for the empirical comparison of the plaintext and ciphertext values showing perfect agreement.
>
> ***there is lacking exploration with respect to non-iid learning approaches.***
>
> We thank the reviewer for this insightful comment. Initially, we observed that pFedMe and Ditto performed well in non-IID settings (cost > 0.3) in the majority of datasets (7/7 and 5/7, respectively). However, your observation prompted a more thorough examination of these algorithms, particularly regarding optimization. In our initial analysis we used a default regularization parameter which controls the strength of personalization vs. aggregation (lower regularization means greater personalization). We have now completed a grid search over the regularization parameter and learning rate and found that non-IID model performance can be improved. While this adjustment does not fully bridge the performance gap in all cases, it reduces it. This led to an interesting discovery: there's a clear correlation between regularization strength and our metric. In IID settings, more regularization (favoring aggregation) performs better while in non-IID settings less regularization (favoring personalization) performs better. While this result seems intuitive, it further highlights the practical utility of our metric as complimentary to these non-IID algorithms. We believe our metric can support regularization parameter tuning for practitioners without incurring the high computational cost of a grid search. We have updated the *Section 7.3* and added the regularization parameter results to the tables in *Section S4.1.2*, accordingly

---

> > ### Comment · Reviewer_8TNG · 2023-12-04
> > **Rebuttal most addressed concerns. Some remaining.**
> >
> > Thank you for the clear rebuttal. I have read it and believe it mostly addresses my concerns. I believe that the better tuning makes the results correctly align and improves the paper. Because of this, I am leaning more positive. However, I keep my score the same due to continued concerns on the privacy preserving ML (ppml) side. In particular, it still remains unclear what the formal privacy preserving guarantee is. Right now, two of the sub protocols use two different ppl techniques (smpc and DP). These provide two different guarantees (confidentiality versus data anonymity). Yet, the final cost metric does not combine them in such a way to provide a meaningful final guarantee. More careful treatment of the ppml guarantee is still needed.

---

### Official Review · Reviewer_qVBz · 2023-11-09

**Soundness:** 2 fair
**Presentation:** 2 fair
**Contribution:** 2 fair
**Rating:** 3
**Confidence:** 2

**Summary:**

The paper proposes a method to calculate dataset similarity which is model agnostic and does not requires any model training. The similarity score can help guide model training with multiple data sources. The paper provide theoretical intuition on the method and presents empirical evidence showing the correlation between the score and utility of model that is trained on multiple datasets.

**Strengths:**

The similarity metric seems to be easy to compute and can be helpful for practitioners who want to train models with multiple data sources.

**Weaknesses:**

1. I'm a bit confused about the relation between data similarity and model utility. Intuitively I think the model utility should be improved the most when we add a dataset that is either too similar (e.g. apparently adding the same dataset would not help at all) or too different (e.g. if we invert all labels, the model might become garbage). But the paper seems to suggest that datasets should be as similar as possible, e.g. in the empirical evaluation Fig 1, and theoretical insights ("When the cosine similarity between x1 and x2 is close to zero, it implies a negligible change in w, leading to minimal improvement in loss"). This is a bit counter-intuitive to me.

2. I think some important concepts and settings need to be explained in more details (which might help resolve my confusions in (1) as well).
a). What is μ and Σ in the algorithm? I guess they are some Gaussian parameters but I don't understand what distribution we're talking about here. (And I presume the "u" in (5) is meant to be "μ"?) In general the intuition of the algorithm is not quite clear to me. I think it might be helpful if the authors can demonstrate a few similarity values for some simple cases, e.g. when D1 and D2 are the same, when one is a subset of the other, or when they're fresh samples drawn from the same distribution etc.
b). In the experiments, how is the data partitioned to form datasets with different level of similarities? I think it's important to examine whether these artificially created datasets reflect the real life scenario where slightly different datasets might be owned by different institutions who want to jointly train model.

**Questions:**

(Those mentioned in the previous question.)

---

> ### Author Response · Authors · 2023-11-16
>
> **Please note: A revised manuscript has been submitted which we reference in our response**
>
> ***the relation between data similarity and model utility....***
>
> We thank the reviewer for their question. When we refer to datasets as 'similar,' we imply that they have similar distribution.  In our results we observe that more similar datasets improve performance and dissimilar datasets worsen performance. This finding aligns with the widely accepted notion that, under the scenario that test-sets are drawn from the same distributions, combining IID datasets improves model performance, but adding non-IID datasets may worsen performance (*Kairouz et al.,arXiv, 2019,  Lin et al., ICML, 2021*).
>
> We also explored scenarios involving datasets with inverted labels or with deliberately contrasting feature distributions per your point. We found that the cost is higher than what we would expect from two random datasets i.e.,> 0.5 and that model performance is no better than chance. We believe this is in keeping with your intuition *‘if we invert all labels, the model might become garbage’*. We have added this to the manuscript *Figure S.5*)
>
> ***and theoretical insights When the cosine similarity between x1 and x2 is close to zero...***
>
> In the theoretical section we are referring to the cosine of the angle between the vectors which ranges from [-1,1], *i.e.,* a score of 0 implies orthogonality. We then use (1-cosine of the angle) in our similarity calculation to ensure the cost is non-negative. We have amended the text to make this clearer.
>
> ***think some important concepts and settings need to be explained in more details...***
>
> We thank the reviewer for this observation. In the algorithm, μ and Σ represent the label mean and covariance matrix. As stated in Section 5. Proposed framework, we model labels as distributions over their features. We have amended the algorithm to clarify what μ and Σ represent. We have also amended u to μ in *eq.5*.
>
> ***demonstrate a few similarity values for some simple cases...***
>
> We thank the reviewer for this question. We use the synthetic dataset to demonstrate various scenarios. As detailed in *Section S.4.1.1* and *Table S.1*, for the synthetic dataset, our approach involves drawing samples from two distinct distributions, varying how much of the two distributions the datasets share. This setup allows us to systematically explore and quantify the similarity under different conditions:
>
> 1. **Datasets drawn from the same distribution:** Both datasets are drawn from the same distribution. We obtain a score of 0.03 (i.e,. highly similar datasets) and find a large improvement in model performance from multi-source learning (MSL).
>
> 2. **Datasets drawn from different distributions**: Each dataset is  drawn from a distinct distribution. We obtain a score 0.5 (i.e., datasets are random to each other) and find that  performance is worse in MSL.
>
> 3. **Identical Datasets**: The datasets are totally identical. We obtain a score of 0  i.e., datasets can be mapped at no cost).
>
> 4. **Dataset is a subset of another**: We subsample (50%) of one dataset to produce the other dataset. We obtain a similarity score of 0.01 i.e., highly similar datasets.
>
> Note points 3 and 4 are new analysis based on your suggestion and have been added to the manuscript in S4.1.1 and Table S.1.
>
> ***how is the data partitioned to form datasets***
>
> To create datasets with varying levels of similarity, we employed different strategies depending on the dataset. *Section S4.1.1* provides a more detailed overview of the approach. As a summary:
>
> 1. **Datasets Without Natural Partition** (Credit, EMNIST, CIFAR-100): For datasets that do not have an inherent or natural partitioning, we implemented label and feature skew as described by *Hsieh et al., ICML, 2019*. These are established techniques for generating non-IID datasets in the federated learning literature.
> 2. **Datasets With Natural Partition (Weather, ISIC, IXITiny)**:
>
>      2a. Weather: We partitioned the data based on climate as described by *Malinin, et al., NeurIPS, 2021*.
>
>      2b. ISIC and IXITiny: For these medical imaging datasets, we partition based on the site and equipment used for data collection as described by du Terrail, Ayed et al., NeurIPS, 2022
>
> ***examine whether these artificially created datasets reflect the real life scenario***
>
> We address this using the ISIC and IXITiny medical imaging datasets, which originate from different institutions and utilize different imaging machines. In *Section 7.2*, we provide a discussion for how our metric captures real-world differences in these datasets and matches intuition on which datasets are likely to be similar. Notably, our analysis with the ISIC dataset reveals an intriguing insight: the type of imaging machine plays a more critical role in dataset similarity than the geographical location of the sites. This finding underscores the practical relevance and applicability of our metric in real-world scenarios.

---

### Author Response · Authors · 2023-11-16
**General comments to reviewers**

**Please note: A revised manuscript has been submitted which we reference in our response**


We thank the reviewers for their constructive feedback. Below we discuss new results or changes that we feel warrant the attention of all reviewers.

**Further exploration of non-IID algorithms**

While we observed that pFedMe and Ditto performed well in non-IID settings (cost > 0.3) in the majority of datasets (7/7 and 5/7, respectively) reviewer comments prompted a more thorough optimization of these algorithms. Specifically, we now conduct a grid search over the learning rate and regularization parameter. Previously we used a default regularization parameter for all datasets, but as this parameter controls the strength of personalization vs. aggregation, it warranted tuning. As expected, we found that model performance  improved performance, especially in non-IID settings. Interestingly, we found a clear correlation between regularization strength and our metric. In IID settings, more regularization (favoring aggregation) performs better while in non-IID settings less regularization (favoring personalization) performs better. Although this result is intuitive, it further highlights the practical utility of our metric. We believe our metric can be used in conjunction with non-IID algorithms by guiding regularization parameter tuning for practitioners without incurring the high computational cost of a hyperparameter grid search. We have updated the discussion in *Section 7.3*, results in *Figure 1* and added the regularization parameter values to the tables in *Section S4.1.2*, accordingly.


**Privacy-preserving methods**

We initially only included a method for calculating feature cost as this required sharing of sample-level information whereas label costs only required sharing of summary statistics. However, we acknowledge that in the low sample size setting, summary statistics can also leak information. We now include a method to calculate label costs in a privacy-preserving manner. To do this, we use a method introduced by *Biswas et al., 2020, NeurIPS *that employs zero-Concentrated Differential Privacy (ρ-zCDP, an intermediate method between pure and approximate DP, see *Section S.1.5* and *eq.S.6*). We chose this method as it is computationally light and has similar sample complexity to non-private methods (particularly relevant in MSL). We show that under a strong privacy guarantee, ρ=0.1 or ε-δ= 0.99,0.01, (*Near and Abuah, Programming Differential Privacy, 2021*) we still retain high accuracy (*Figure S.2*). Note, we chose ourε-δ values based on the suggestions in the literature to provide privacy guarantees (*Ponomareva et al., 2023, J.Artif.Intell.Res*). We have amended *Algorithm 1* and added *Section S.2.3* describing the method in more detail. We believe now the metric can be fully calculated in a privacy-preserving way and, importantly, this computation can be efficiently executed in parallel. Note, to reviewer 8TNG, we address your comments on security guarantees separately in our direct response.


**Motivating the Hellinger distance**

Our primary reason for using the Hellinger distance was to introduce a bounded cost function that is admissible in the Wasserstein framework. Specifically, the Wasserstein distance is a special case of the (f, Γ)-divergence family where the Γ cost function is bounded 1-Lipschitz functions (e.g., L1 or L2 norm) (*Birrell et al., JMLR, 2022*). However, it is possible to substitute it with the Hellinger distance as it retains the necessary properties. The advantage of this modification is that it shifts interpretation from assessing distribution differences in Euclidean geometry to comparing probability densities within the manifold. This is because the Hellinger distance can be interpreted as representing the shortest geodesic distance in the statistical manifold. This approach provides a way to compare label distributions that preserve the optimal transport interpretation, while ensuring boundedness and relevance in probability space. We now include this more detailed rationale for Hellinger distance in *Section 6.2*.


**Correlation to gradient diversity**

We empirically test the correlation between gradient diversity and our metric. As a comparison, we also test the correlation between gradient diversity and standard Wasserstein distance. Our findings reveal that our metric consistently exhibits a higher correlation with gradient diversity compared to the Wasserstein distance across all datasets (with a statistically significant paired t-test, p=0.028, see *Table S.9*). We believe this can be attributed to the choice of cosine similarity-hellinger distance that more closely aligns with the dynamics of neural network training.

---

### Meta-Review · Area_Chair_dVNX · 2023-12-08

**Metareview:**

Unfortunately, reviewers were not supportive of the paper, even after the rebuttal. Couple of main concerns that were prominent in the reviews were: a) the lack of rigor in defining the privacy protection (see comments from reviewer 8TNG), and b) lack of explanation of the key concepts in the paper (see comments from reviewer qVBZ). In light of that, we would recommend rejecting the paper, and encourage the authors to submit to a different venue after incorporating the comments.

**Justification For Why Not Higher Score:**

Reviewers were not supportive of the paper (including the expert reviewers).

**Justification For Why Not Lower Score:**

NA

---

### Decision · Program_Chairs · 2024-01-16

Reject